



# A typical weather pattern for the ozone pollution events in North China

Cheng Gong[1,2], Hong Liao[3*]

[1]State Key Laboratory of Atmospheric Boundary Layer Physics and Atmospheric Chemistry (LAPC), Institute of

Atmospheric Physics, Chinese Academy of Sciences, Beijing, 100029, China,

[2]University of Chinese Academy of Sciences, Beijing, 100029, China,

[3]Jiangsu Key Laboratory of Atmospheric Environment Monitoring and Pollution Control, Jiangsu Collaborative Innovation Center of Atmospheric Environment and Equipment Technology, School of Environmental Science and Engineering, Nanjing University of Information Science and Technology, Nanjing, 210044, China

*Correspondence to*: Hong Liao (hongliao@nuist.edu.cn)

**Abstract.** Ground-level observations, reanalyzed meteorological fields and a 3-D global chemical and transport model (GEOS-Chem) were applied in this study to investigate ozone ($O_3$) pollution events (OPEs) in North China (36.5°N-40.5°N, 114.5°E-119.5°E) during 2014-2017. Ozone pollution days (OPDs) were defined as days with maximum daily averaged 8-h (MDA8) concentrations over North China larger than 160 μg m$^{-3}$, and OPEs were defined as periods with 3 or more

consecutive OPDs. Observations showed that there were 167 OPDs and 27 OPEs in North China during 2014-2017, in which 123 OPDs and 21 OPEs occurred in May-July. We found that OPEs in North China occurred under a typical weather pattern with high daily maximum temperature (Tmax), low relative humidity (RH), anomalous southerlies and divergence in the lower troposphere, an anomalous high-pressure system at 500 hPa and an anomalous downward air flow from 500 hPa to the surface. Under such a weather pattern, chemical production of $O_3$ was high between 800 and 900 hPa, which was then

transported downward to enhance $O_3$ pollution at the surface. A standardized index I_OPE was defined by applying four key meteorological parameters, including Tmax, RH, meridional winds at 850 hPa (V850) and zonal winds at 500 hPa (U500). I_OPE can capture approximately 80 % of the observed OPDs and OPEs, which has implications for forecasting OPEs in North China.

## 1 Introduction

Ground-level ozone ($O_3$) is generated by photochemical reactions involving nitrogen oxides ($NO_x$) and volatile organic compounds (VOCs) (FinlaysonPitts and Pitts, 1997; Sillman, 1999). Enhanced surface $O_3$ concentrations increase premature mortality (e.g., Bell et al., 2006; Anenberg et al., 2010; Lelieveld et al., 2015; Nuvolone et al., 2018) and reduce crop yields (e.g., Fuhrer et al., 1997; Krupa et al., 1998; Ainsworth et al., 2012; Mills et al., 2018). $O_3$ pollution events (OPEs) occur frequently in megacities with sufficient $O_3$ precursors during summertime when solar radiation is strong (Solomon et al.,



2000; Wang et al., 2006a; Wang et al., 2006b; Roy et al., 2008; Carro-Calvo et al., 2017; Fix et al., 2018). As a result, the formation mechanisms of and prevention strategies for ground-level $O_3$ has been a focus in many countries around the world.

Ozone concentrations are influenced by meteorological parameters. High temperature can change $O_3$ concentrations by accelerating $O_3$ chemical production rates and enhancing natural emissions such as biogenic emissions and $NO_x$ from soil

(Jacob and Winner, 2009). Blommer et al. (2009) analyzed observed $O_3$ from 1987 to 2007 across the rural eastern US and showed that as temperature increased by 1 K, $O_3$ concentrations increased by an average of 3.2 ppbv prior to 2002 but increased by an average of 2.2 ppbv after 2002 because of the reduction in anthropogenic $NO_x$ emissions. Rasmussen et al. (2012) used observed $O_3$ and temperature in the eastern US during 1988-2009 to characterize the sensitivity of summer time $O_3$ to temperature. These authors showed that the sensitivities were 3-6 ppbv $K^{-1}$ over the northeast, 3-4 ppbv $K^{-1}$ over the

Great Lakes, and 3-6 ppbv $K^{-1}$ over the Middle Atlantic states. Relative humidity (RH) is also found to be an important parameter for $O_3$ formation. Zhang et al. (2015) showed that days with the highest 10 % $O_3$ concentrations were associated with lower RH than days with the lowest 10 % $O_3$ concentrations in Guangzhou by examining continuous observations of $O_3$ and meteorological parameters during March 2013 to February 2014. Kavassalis and Murphy (2017) reported a negative correlation between summer-time $O_3$ concentrations and RH on the basis of observed $O_3$ and RH from 1987 to 2015 at 101

rural sites in the US. Moreover, cloud fraction influences $O_3$ concentrations by changing the near-surface solar radiation and hence photochemical reaction rates. Jeong and Park (2013) showed, by using a 3-D global chemical and transport model (GEOS-Chem), that the increases in $O_3$ concentration in East Asia from 1985-1989 to 2002-2006 could be explained in part by the decreases in cloud cover.

In addition to the local meteorological parameters, $O_3$ concentrations are also influenced dynamically by large-scale

circulations. By analyzing 11 years of ozonesonde data, Zhou et al. (2013) showed that the interannual variability of $O_3$ over Hong Kong was closely associated with the East Asian monsoon; circulations during monsoon season influence the transport of continental pollutants to Hong Kong. Liao et al. (2017) carried out composite analysis on observed surface $O_3$ concentrations in the Yangtze River Delta (YRD) during 2013-2016 for ten typical circulation types identified by the automated Lamb weather type approach (Jenkinson and Collison, 1977). These authors found that $O_3$ concentrations in the

YRD were high under the influence of westerlies, which occur frequently in summer associated with the subtropical high. Under such conditions, high temperatures and strong solar radiation in the YRD, together with the transport of biogenic VOCs from the mountain areas of Anhui and Zhejiang provinces, led to high $O_3$ levels in the YRD. Zhao and Wang (2017) reported that the daily variability of West Pacific subtropical high (WPSH) can influence the daily variability of surface-layer $O_3$ over eastern China in summer of 2014-2016. They found, by using observed $O_3$ and reanalyzed data, that $O_3$

concentrations decreased in South China and increased in North China during days with a high WPSH-I index, which is an indicator of the intensity of WPSH at the 500 hPa level. A strong WPSH leads to moist, cloudy weather and low temperatures in South China and dry, sunny weather in North China.

Previous studies also reported that OPEs are influenced by meteorological conditions. Zhang et al. (2017), utilizing 30 years of $O_3$ observations and meteorological variables over the US, showed that $O_3$ extreme days (location-specific 95th percentile)





overlapped with 32 % of temperature extreme days, along with low RH and low wind speed. By using both observations and a regional chemistry-climate model, Pu et al. (2017) showed that a heat wave event in YRD during the summer of 2013 led to a severe $O_3$ pollution episode with a peak $O_3$ concentration of 160.5 ppbv as a result of the accelerated chemical reaction, low cloud fraction and stagnant conditions. By using the GEOS-Chem model and observed $O_3$ concentrations, Zhang and

Wang (2016) showed that extreme drought events also led to three high $O_3$ episodes (with peak concentrations of 70 ppbv) in October 2010 in the southeast US by the enhanced emissions of biogenic isoprene from water-stressed plants. Moreover, regional transport of $O_3$ and precursors (such as $NO_x$ mand isoprene) are important for OPEs. For example, Whaley et al. (2015) used the GEOS-Chem model with tagged-$O_3$ to identify the sources of $O_3$ for 15 OPEs in Toronto during 2004-2007 and found that $O_3$ in the northeast US contributed 26 % to $O_3$ in Toronto during OPEs. They also used the GEOS-Chem

adjoint model to examine the sensitivities of $O_3$ concentrations during OPEs in Toronto to emissions of precursors in different regions and found a strong sensitivity to the southern Ontario and US fossil fuel $NO_x$ emissions and natural isoprene emissions. Currently, previous studies on OPEs in China were focused on one single observational site or a few episodes (e.g., Wang et al., 2006c; Shen et al., 2015; Li et al., 2017a), and few studies have systematically examined OPEs in a regional scope, especially for North China (36.5 °N-40.5 °N, 114.5 °E-119.5 °E), where the highest $O_3$ peak concentrations

were observed (Wang et al., 2017).

The scientific goals of this work are as follows: (1) to characterize the frequencies and intensities of OPEs in North China, (2) to identify key meteorological parameters that can be used to define a typical weather pattern for OPEs in North China, and (3) to quantify the contributions of different chemical and physical processes to OPEs under such a typical weather pattern. The integrated process rate (IPR) analysis is a widely used method to quantify the contributions of different processes to $O_3$

(Goncalves et al., 2009; Jiang et al., 2012; Li et al., 2012). In Sect. 2, observed $O_3$ concentrations, reanalyzed meteorological data, a model description, and the IPR analysis method are briefly introduced. Section 3 presents the observed and spatiotemporal distributions of OPEs in North China during 2014 to 2017. Section 4 describes the key meteorological parameters that lead to OPEs and the definition of a standardized index to represent a typical weather pattern for OPEs. Section 5 examines how the typical weather pattern leads to OPEs by IPR analysis in the GEOS-Chem model.

**2 Methods**

**2.1 Observed ground-level $O_3$ concentrations**

The ground-level hourly $O_3$ concentrations are obtained from the national air quality monitoring network of China (http://datacenter.mep.gov.cn/websjzx/queryIndex.vm), which was established in 2012 by the Ministry of Environment Protection of China. $O_3$ concentrations from this network have units of μg m$^{-3}$. Under the condition of 25 °C and 1013.25 hPa,

1 μg m$^{-3}$ of $O_3$ is approximately 0.5 ppbv. Hourly $O_3$ concentrations are available at 1582 sites during 2014-2017. For each site, the maximum daily 8-h average concentration (MDA8) of $O_3$ is calculated by utilizing an 8-h moving average window for each day. To ensure the data quality, the 8-h moving window has to contain more than 6-h valid observations, and the





number of days with valid O₃ MDA8 has to be more than 15 for each month. As a result, 740 among the 1582 sites in China (67 sites among the 114 sites in North China ($36°\text{-}40.5°N$, $114.5°\text{-}119.5°E$)) are selected and used in this study. The spatial distribution of these selected sites and the region of North China are shown in Fig. 1.

The China National Ambient Air Quality Standard (GB3095-2012) states that $O_3$ concentration exceeds the national air quality standard if the MDA8 O3 concentration of a location is higher than 160 μg m⁻³. In this study, we aim to investigate $O_3$ pollution over a large area rather than at a single site; we define $O_3$ polluted days in North China as the days with MDA8 $O_3$ concentrations averaged over North China exceeding 160 μg m⁻³. We also define an ozone episode in North China as three or more consecutive days of regional $O_3$ pollution.

## 2.2 Reanalyzed meteorological fields

Meteorological fields are taken from Version 2 of Modern Era Retrospective-analysis for Research and Application (MERRA2), which was generated from the NASA Global Modeling and Assimilation Office (GMAO) by using Version 5 data assimilation system (DAS) of the Goddard Earth Observing System Model. Compared with the first version of MERRA, MERRA2 has assimilated more observations and made many improvements and updates in DAS (Molod et al., 2015). The MERRA2 reanalyzed meteorological dataset in the Extended Asia domain ($11°S\text{-}55°N$, $60°E\text{-}150°E$) has a horizontal resolution of $0.5°$ latitude $\times$ $0.667°$ longitude and 47 vertical layers up to 0.01 hPa. The temporal resolution for surface meteorological parameters (such as 2-meter air temperature) is 1 h and that for atmospheric meteorological parameters (such as relative humidity and wind) is 3 h. To investigate the key meteorological factors that lead to OPEs, daily maximum 2-meter temperature (Tmax), daily mean relative humidity (RH) at the surface, daily averaged meridional and zonal winds at 850 hPa and 500 hPa (U850, V850, U500 and V500, where westerlies and southerlies have positive values) during 2014-2017 are utilized. In addition, due to the lack of geopotential heights in the MERRA2 dataset, daily mean geopotential heights at 850 hPa and 500 hPa from the National Center for Environmental Prediction (NCEP) and National Center for Atmospheric Research (NCAR) global reanalysis at a resolution of $2.5°$ latitude by $2.5°$ longitude are utilized. All the time series of meteorological parameters have been detrended first and then standardized by their respective standard deviation to remove interannual or seasonal variability.

## 2.3 GEOS-Chem model

The hourly $O_3$ concentrations from May to July for 2014-2017 are simulated by the nested version of the 3-D global chemical transport model (GEOS-Chem, version 11-01) driven by the MERRA2 reanalysis meteorological data. Over the nested domain ($11°S\text{-}55°N$, $60°E\text{-}150°E$), the model resolution is the same as that of the MERRA2 dataset, as described above. Concentrations of all tracers in lateral boundaries are provided by the global GEOS-Chem simulation with $2°$ latitude $\times 2.5°$ longitude horizontal resolution.





The GEOS-Chem model employs a fully coupled $NO_x$-$O_x$-hydrocarbon-aerosol chemistry mechanism (Bey et al., 2001; Park et al., 2003; Pye et al., 2009) to simulate concentrations of gas-phase pollutants (such as $NO_x$ and $O_3$) and aerosols (including sulfate, nitrate, ammonium, OC and BC, sea salt, and mineral dust). The LINOZ scheme is used for stratospheric $O_3$ chemistry (McLinden et al., 2000). The vertical mixing in planetary boundary layers (PBL) is calculated by a nonlocal

scheme (Lin and McElroy, 2010). The anthropogenic emissions of CO, $SO_2$, $NO_x$, $NH_3$ and VOCs in the simulated domain are obtained from MEIC emission inventory, which includes emissions from industry, power, residential and transportation sectors from 2014 to 2017 (Li et al., 2017b). The biogenic emissions in GEOS-Chem employ the MEGAN v2.1 biogenic emissions with updates from Guenther et al. (2012).

## 2.4 IPR analysis method

Five major processes that influence $O_3$ concentrations include net chemical production, horizontal advection, vertical advection, dry deposition, and diffusion (vertical PBL mixing process in GEOS-Chem model). Integrated process rate (IPR) analysis is used to evaluate the daily relative contributions of individual processes to an OPE in the studied domain by using the following formula (Goncalves et al., 2009):

$$PC_i(\%) = \frac{PC_i}{\sum_i^n abs(PC_i)} \times 100 \%,$$    (1)

where $PC_i$ is the percentage contribution of process $i$ to $O_3$ mass in the specific domain and $abs(PC_i)$ is the absolute value of $PC_i$. $n$ is the total number of processes ($n$ is 5 in our analysis). $PC_i(\%)$ is the relative contribution of process $i$ to $O_3$ mass. It is noted that the sum of process contributions ($PC_i(\%)$) is not 100 %, but the sum of the absolute values of $PC_i(\%)$ equals 100 %. The IPR analysis method has been applied to identify the key processes contributing to extreme air pollution episodes as well as the interannual and decadal variations (Mu and Liao, 2014; Lou et al., 2015; Shu et al., 2016).

## 3 Frequencies and intensities of OPEs in North China

### 3.1 Spatiotemporal distributions of surface layer $O_3$

Figure 1 shows the monthly mean MDA8 $O_3$ concentrations averaged over 2014-2017 at the 740 observational sites. The MDA8 $O_3$ values show obvious seasonal variations in eastern China. The monthly mean MDA8 $O_3$ values at most sites in eastern China were lower than 100 μg m$^{-3}$ during November to March, while the values were generally high during April-

October, especially in North China and the YRD region. North China had the highest MDA8 $O_3$ concentrations from May to July. In June, the most polluted month, the MDA8 $O_3$ concentrations at 40 % (25/62) of observational sites in North China exceeded 160 μg m$^{-3}$, in which four sites (two sites in Baoding, one in Hengshui and the other in Zibo) even exceeded 180 μg m$^{-3}$.





Figure 2a shows the seasonal and interannual variations in MDA8 $O_3$ concentrations averaged over all 62 sites in North China. The MDA8 $O_3$ concentrations in North China peaked in June and had relatively high values from May to July. In 2016 and 2017, a secondary peak of concentration showed up in September, but it is difficult to conclude whether this was a general or accidental feature with the limited four years of data. With respect to the interannual variation, MDA8 $O_3$ concentrations in most months exhibited an increasing trend from 2014 to 2017. The MDA8 $O_3$ concentration over North China reached the highest value of 182 μg m$^{-3}$ in June of 2017. This increasing trend indicates that the strict emission reduction measures in China in recent years had little effect on $O_3$ pollution in North China.

### 3.2 Ozone polluted days and the frequency of OPEs

Figure 2b shows the $O_3$ polluted days in North China (the days with an average MDA8 $O_3$ concentration over North China exceeding 160 μg m$^{-3}$) in different months of 2014-2017. From 2014 to 2017, there were 167 $O_3$ polluted days in North China, in which 123 days (70 %) occurred in the months of May to July. In 2014, July and August had the highest number of $O_3$ polluted days (10 days). In 2015-2017, the number of $O_3$ polluted days was the highest in June and kept increasing. Ozone polluted days in North China had values of 11, 16 and 20 days in June of 2015, 2016, and 2017, respectively.

Figure 2c shows the number of OPEs in North China in each month of 2014-2017. An $O_3$ pollution event in North China is defined as three or more consecutive days of $O_3$ pollution. There were 27 OPEs in the studied time period, and 21 of these OPEs occurred in May to July. Except for June of 2014, North China suffered 1−3 OPEs per month in May to July of 2014-2017. As shown above, $O_3$ pollution in North China was the worst in May to July. The 21 OPEs in these three months of 2014-2017 are further analyzed in the following sections.

### 3.3 Intensities of OPEs in North China

Figure 3 shows the mean and maximum MDA8 $O_3$ concentrations as well as the duration of 21 OPEs over May to July in the years of 2014-2017 in North China. The averaged MDA8 $O_3$ concentration for OPEs is 193.0 μg m$^{-3}$, indicating high intensities of OPEs. The maximum MDA8 $O_3$ concentrations for a single day during OPEs can even reach 243.8 μg m$^{-3}$, and over half of the episodes (11/21) have at least one day where MDA8 $O_3$ concentrations exceed 200 μg m$^{-3}$. Moreover, OPEs last for many consecutive days. The mean duration of OPEs is 4.3 days, while some episodes can last for one week and even longer (e.g., the episodes starting from June 16$^{th}$, 2016 and June 14$^{th}$, 2017). Understanding the kind of weather pattern that leads to these long-lasting OPEs with high $O_3$ concentrations is quite necessary.


## 4 A Typical weather pattern for OPEs

### 4.1 Composited weather pattern for OPEs

Figure 4 shows the composited weather pattern for 21 OPEs identified in North China (36.5 °N-40.5 °N, 114.5 °E-119.5 °E) during May to July of 2014-2017. We examine the composited Tmax, RH, winds and SLP at the surface, winds and geopotential height at 850 hPa, winds and geopotential height at 500 hPa, vertical pressure velocity and divergence. All these daily parameters in May to July of 2014-2017 are detrended first to remove interannual or seasonal variability. Then, for each parameter, we calculate the standardized anomalies by using the following formula:

$$[x_{i,d}] = \frac{x_{i,d} - \overline{x_i}}{s_i}, \tag{2}$$

where $x_{i,d}$ indicates the detrended value for parameter $i$ on day. $\overline{x_i}$ and $s_i$ indicate the mean value and the standard deviation of the detrended daily time series for parameter $i$, respectively. $[x_{i,d}]$ is the standardized anomaly for parameter $i$ on day $d$. During OPEs, positive Tmax anomalies (Fig. 4d) and negative RH anomalies (Fig. 4e) occur in North China, indicating hot and dry weather conditions at the surface. The wind and pressure fields show a similar pattern at the surface (Fig. 4c) and at 850 hPa (Fig. 4b). Anomalous southerlies prevail in North China, accompanied by anomalous high pressure in the east and anomalous low pressure in the west. At the 500 hPa altitude, North China is under the influence of an anomalous anti-cyclone (high pressure) (Fig. 4a), which causes high temperature and low RH at the surface.

The composited pressure-latitude cross-sections of vertical velocity and divergence for 21 OPEs from 1000 hPa to 500 hPa averaged over 114.5 °E to 119.5 °E, the west and east boundary of North China, are shown in Fig. 4f and 4g, respectively. Except for the north of 39 °N under 850 hPa, North China shows a downward airflow anomaly from 1000 hPa to 500 hPa during OPEs, which is a typical feature of the high-pressure system. In fact, the upward anomaly under 850 hPa in the northern domain is a fake signal because the elevation sharply increases to approximately 1000 m at Yan Mountain to the north of 39 °N, which leads to the surface pressure being lower than 900 hPa (~1000 m) or even 850 hPa (~1500 m). As a result, the vertical velocity under 850 hPa for the reanalyzed dataset is unreliable to the north of 39 °N. Figure 4g shows the divergence anomaly during OPEs in North China. Strong divergence occurs between 950 and 850 hPa. The anomalous downward flow transports air to the lower troposphere and leads to the anomalous divergence.

### 4.2 Correlations between meteorological parameters and O$_3$ concentrations

To identify the key meteorological factors associated with the MDA8 O$_3$ concentrations in North China, we examine the correlation coefficients between the MDA8 O$_3$ concentration averaged over North China and the meteorological parameters, including daily Tmax and daily mean RH, planetary boundary layer height (PBLH), surface level pressure (SLP), and meridional and zonal wind speed at 1000 hPa (U1000, V1000), 850 hPa (U850, V850) and 500 hPa (U500, V500). These parameters at each grid cell are detrended first and then standardized as described in Sect. 4.1. Figure 5 shows the correlation coefficients between daily MDA8 O$_3$ concentration in North China and the ten standardized meteorological parameters.





MDA8 $O_3$ concentrations in North China exhibit positive correlation with Tmax (Fig. 5a), PBLH (Fig. 5c), V1000 (Fig. 5f) and V850 (Fig. 5h) in the vicinity of North China and with U500 (Fig. 5i) in the north and V500 (Fig. 5j) in the west of North China. MDA8 $O_3$ in North China has a negative correlation with RH (Fig. 5b), SLP (Fig. 5d), U500 (Fig. 5i), and V500 (Fig. 5j). MDA8 $O_3$ is found to have a weak correlation with U1000 (Fig. 5e) and U850 (Fig. 5g). It should be noted

that some meteorological factors are closely related. For instance, previous studies have revealed that PBLH is positively correlated with surface temperature (Zhang et al., 2013) but negatively correlated with SLP (Seidel et al., 2010; Guo et al., 2016). Winds at 1000 hPa and 850 hPa are usually highly correlated and show similar patterns. As a result, four meteorological factors are selected to represent the key meteorological conditions for high MDA8 $O_3$ concentrations: Tmax represents the thermal condition, RH indicates the humidity condition, 850 hPa zonal winds indicate circulation in the lower

atmosphere and 500 hPa meridional winds describe the dominate large-scale circulation.

**4.3 Definition of I_OPE**

As described above, the weather pattern associated with high MDA8 $O_3$ concentrations in North China can be characterized by high Tmax and low RH at the surface, anomalous southerlies in the lower atmosphere, and anomalous high pressure at the 500 hPa level. We can then define an index I_OPE to represent such a weather pattern and to examine how many $O_3$ polluted

days and OPEs in North China occurred under such a weather pattern. For a specific day, I_OPE is defined as follows:

$$I\_OPE = [\textstyle\sum_x index\_x], \tag{3}$$

where $x$ indicates Tmax, RH, V850 or U500, and the square bracket indicates standardization. The four index_x values are calculated by:

$$index\_Tmax = \left[\textstyle\sum_{i,j}^{35°N-45°N,110°E-120°E} Tmax_{i,j}\right], \tag{4}$$

$$index\_RH = -\left[\textstyle\sum_{i,j}^{35°N-45°N,110°E-120°E} RH_{i,j}\right], \tag{5}$$

$$index\_V850 = \left[\textstyle\sum_{i,j}^{35°N-45°N,107°E-120°E} V850_{i,j}\right], \tag{6}$$

$$index\_U500 = \left[\textstyle\sum_{i,j}^{45°N-55°N,105°E-125°E} U500_{i,j}\right] - \left[\textstyle\sum_{i,j}^{34°N-40°N,105°E-125°E} U500_{i,j}\right], \tag{7}$$

where $i$ and $j$ indicate latitude and longitude of the grid cell, respectively. $Tmax_{i,j}$, for example, is the Tmax in grid $(i, j)$ on a specific day after the time series is detrended and standardized, as described in Sect. 4.1. Domains with strong correlation

between each parameter (Tmax, RH, V850, or U500) and MDA8 $O_3$ concentrations in North China are shown in Fig. 5 by red rectangles. Since RH exhibits a negative correlation with MDA8 $O_3$ concentrations, index_RH has a negative sign in Eq. (5).

Figure 6 shows the time series of MDA8 $O_3$ concentrations in North China and the five indexes (index_Tmax, index_RH, index_V850, index_U500 and I_OPE). The OPEs in Fig. 6a (pink rectangles) are captured by the five indexes. Among all



the indexes, index_Tmax has the strongest correlation with MDA8 $O_3$ concentrations, with a correlation coefficient ($r$) of 0.52. The correlation coefficients between index_RH, index_V850 and index_U500 and MDA8 $O_3$ concentrations are 0.32, 0.35, and 0.41, respectively. It is interesting that the correlation coefficient between I_OPE and MDA8 $O_3$ is 0.64, which is higher than that between each individual index and MDA8 $O_3$, indicating that MDA8 $O_3$ concentrations in North China are

influenced by multiple meteorological factors rather than a single factor.

Figure 7 shows the composite patterns of anomalies of meteorological fields (Tmax at the surface, RH at the surface, winds at 850 hPa, and winds at 500 hPa) for OPEs (Fig. 7a-d) and for days with I_OPE >0 (Fig. 7e-h). The similarity in patterns between these two types of composite analyses indicates that I_OPE can capture the weather pattern associated with OPEs, including the high Tmax, low RH, anomalous southerlies at 850 hPa, and the anomalous anti-cyclonic winds at 500 hPa.

From May to July over 2014-2017, there were 123 days with regionally averaged observed $O_3$ concentrations of greater than 160 μg m$^{-3}$, in which 82.1 % days (101/123) occurred under the condition of I_OPE>0. Conversely, 54.3 % days (101/186) with I_OPE>0 and 72.5 % days (37/51) with I_OPE>1 were observed for $O_3$ pollution days in North China. Among the observed 21 OPEs (90 days), 17 OPEs (69 days) occurred under a weather pattern with I_OPE>0. Therefore, I_OPE can be used as a meteorological predictor for OPEs in North China.

**5 Simulated OPEs and IPR analysis**

**5.1 Simulated OPEs**

We have identified a typical weather pattern associated with OPEs in North China, as presented in Sect. 4. Here, we use the GEOS-Chem simulation of $O_3$ in May-July of 2014-2017 to quantify the contributions of different chemical and physical processes to OPEs under such a weather pattern. Figure 8 shows the time series of observed and simulated daily MDA8 $O_3$

averaged over North China. The correlation coefficients between the observed and simulated MDA8 $O_3$ are 0.53, 0.64, 0.61, and 0.71 in 2014, 2015, 2016, and 2017, respectively, indicating that the GEOS-Chem model can simulate the daily variation in MDA8 $O_3$. Compared to observed MDA8 $O_3$ concentrations, the simulated concentrations have a mean bias (MB) (normalized mean bias (NMB)) of 2.4 μg m$^{-3}$ (1.7 %) in 2014, 6.7 μg m$^{-3}$ (4.9 %) in 2015, 1.8 μg m$^{-3}$ (1.2 %) in 2016, and - 12.5 μg m$^{-3}$ (-8.4 %) in 2017. For all the data samples in May-July of 2014-2017, the observed MDA8 $O_3$ concentration

averaged over North China is 146.8 μg m$^{-3}$, and the simulated mean value is also 146.8 μg m$^{-3}$. The linear regression through the origin between observed and simulated MDA8 $O_3$ has a regression coefficient of 0.96, indicating the capability of the model in simulating the MDA8 $O_3$ concentrations.

The GEOS-Chem model, however, has some difficulties in capturing the peak values of MDA8 $O_3$, as reported in previous studies by Zhang and Wang (2016) and Ni et al. (2018). During May-July of 2014-2017, for the $O_3$ polluted days with

observed MDA8 $O_3$ > 160 μg m$^{-3}$, comparisons of simulated values with observations show an NMB of -14.6 %. As a result, if the same threshold (160 μg m$^{-3}$) is applied in the model to define $O_3$ polluted days, only 8 OPEs (highlighted by pink rectangles in Fig. 8) among the 17 OPEs with I_OPE>0 can be captured by the model. Considering that the model has an



NMB of -14.6 % for the days with observed MDA8 $O_3 > 160$ µg m$^{-3}$, a revised lower threshold of 136.6 µg m$^{-3}$ (160*85.4 %) is adopted to define the $O_3$ polluted days in the model, and consequently, 6 more OPEs are identified (highlighted by light blue rectangles in Fig. 8). Therefore, among the 17 OPEs (69 days) under the typical weather pattern, 14 episodes (59 days) can be identified by the model. We then carry out IPR analysis for these 14 episodes (59 days) to understand how the typical

weather pattern leads to OPEs in North China.

## 5.2 IPR analysis

Five processes that influence $O_3$ concentrations are analyzed, including net chemical production, horizontal advection, vertical advection, dry deposition, and diffusion. Note that wet deposition is not considered because of its small contribution to $O_3$ budget (Mickley et al., 1999; Liao et al., 2006). All of the processes are diagnosed at every time step and then summed

over each day in the simulation. To avoid the discrepancy in $O_3$ budget because of different lasting days of OPEs, the daily mean $O_3$ mass flux (MF (Gg $O_3$ day$^{-1}$)) is presented for each process. We also calculate PC(%), as described in Sect. 2.4, to examine the relative percentage contribution of each process. The horizontal domain for the IPR analysis is North China (36.5°N-40.5°N, 114.5°E-119.5°E). We diagnose first the vertical profiles of MF for the model layers from the surface to 500 hPa averaged over all days in May-July of 2014-2017 and then quantify the anomalies in MF during OPEs relative to the

seasonal mean flux to identify the major changes in processes that lead to high $O_3$ episodes. Finally, mechanisms that lead to OPEs in North China are discussed on the basis of process analysis.

## 5.2.1 Vertical profiles of MF averaged over May-July of 2014-2017

Figure 9a shows the vertical profiles of MF for each process over North China averaged over all days in May-July of 2014-2017. Note that the MF of each process at a specific level indicates the net $O_3$ mass change within this level rather than the

flux across this level, especially for the vertical processes such as diffusion and vertical advection.

Net chemical production at the surface is a large negative value (-2.5 Gg $O_3$ day$^{-1}$) (Fig. 9a) as a result of the $O_3$ titration effect by high $NO_x$ concentrations at the surface. In the upper layers, because of the decreases in $NO_x$ concentrations and the stronger radiations, net chemical production has positive contributions to $O_3$ concentration over North China, with high values exceeding 1.4 Gg $O_3$ day$^{-1}$ at approximately 900 hPa and 800 hPa. Note that net chemical production is practically the

only process that increases $O_3$ between 930 hPa and 800 hPa. Above 750 hPa, net chemical production decreases due to the decreases in $O_3$ precursors.

Diffusion is a process that mainly occurs in the boundary layer, which transports $O_3$ along the concentration gradient and mixes $O_3$ evenly. The sum of the mass fluxes of diffusion from the surface to 850 hPa is small (0.36 Gg $O_3$ day$^{-1}$), indicating that diffusion has a small effect on the total mass of $O_3$ in the boundary layer. However, the diffusion process is important in

the boundary layer, which has negative contributions between 950 and 800 hPa but a positive contribution at the surface (Fig. 9a). Since the mass flux of diffusion for the whole boundary layer is small, it is indicated that $O_3$ aloft is transported





downward to be mixed at the surface by the diffusion process. Such downward transport and mixing of $O_3$ were also reported in previous IPR analyses (e.g., Khiem et al., 2010; Li et al., 2012; Tang et al., 2017).

Vertical advection exhibits negative MF values from the surface to approximately 750 hPa and then becomes positive in the upper layers (Fig. 9a), indicating that $O_3$ is transported from the lower to upper atmosphere by vertical advection under the

seasonal mean condition. Horizontal advection increases $O_3$ from the surface to approximately 900 hPa but decreases $O_3$ at the upper levels (Fig. 9a). Dry deposition occurs at the surface and has an MF of -4.9 Gg $O_3$ day$^{-1}$ under the seasonal mean condition.

## 5.2.2 Comparison of processes during OPEs with the seasonal mean values

Figure 9b shows the profiles of the anomaly of each process during OPEs relative to the seasonal mean value over May-July

of 2014-2017. During OPEs, net chemical production at layers from the surface to approximately 800 hPa is enhanced significantly, generating $O_3$ in North China. The largest enhancement occurs between 950 hPa and 800 hPa, exceeding +0.3 Gg $O_3$ day$^{-1}$. With respect to diffusion during OPEs, both the positive contribution at the surface and the negative contributions in the upper layers increase (Fig. 9b), indicating that more $O_3$ is mixed from the upper levels to the surface to increase the surface $O_3$ concentration during OPEs.

The vertical and horizontal advections during OPEs are the processes that have the largest changes relative to the mean condition. Anomalous vertical advection increases $O_3$ from the surface to approximately 800 hPa but decreases $O_3$ above 700 hPa. A large amount of $O_3$ is transported from aloft to the lower atmosphere by vertical advection, which will be examined in detail in Sect. 5.2.3 below. Horizontal advection reduces $O_3$ from the surface to approximately 800 hPa, which will also be explained in Sect. 5.2.3.

Since $O_3$ concentrations at the surface are determined by the processes in the boundary layer, we show in Table 1 the seasonal mean MF, the absolute MF during OPEs, and their difference for each process in the boundary layer (from the surface to 850 hPa) over North China. Relative to the mean condition, net chemical production, diffusion, dry deposition, horizontal advection, and vertical advection during OPEs change by 3.3, -1.2, -0.4, -11.4, and 10.4 Gg $O_3$ day$^{-1}$, indicating that net chemical production, horizontal advection, and vertical advection are the most dominant processes that lead to OPEs.

During OPEs, net chemical production and vertical advection increase $O_3$ in North China, while horizontal advection reduces $O_3$ in this region.

## 5.2.3 Mechanisms for the typical weather pattern leading to OPEs

The typical weather pattern for OPEs in North China has been identified in Sect. 4, which is characterized by hot and dry air at the surface, anomalous southerlies and divergence in the lower troposphere, anomalous high pressure at 500 hPa and

anomalous downward airflows from 500 hPa to the surface. The hot and dry air under the high-pressure system accelerates chemical production of $O_3$ in the boundary layer (e.g., Zhang and Wang, 2016; Pu et al., 2017). Moreover, the hot air is



beneficial for developing the mixed layer, leading to more $O_3$ mixed downward to the surface during OPEs, as described in Sect. 5.2.2.

The diagnosed vertical advection anomaly during OPEs can be explained by Fig. 10a, which shows the pressure-latitude cross-section of simulated daily mean $O_3$ concentrations as well as the anomalous vertical pressure velocity profile averaged

over North China during OPEs. Note that the regional mean vertical velocity near the surface is interfered by the Yan Mountain, as described in Sect. 4.1, so we will not discuss the vertical air flow below 950 hPa. The anomalous downward air flow is high at 850 hPa, and the net chemical production of $O_3$ is still strong above 850 hPa (Fig. 9b), leading to the large transport of $O_3$ to the boundary layer to form OPEs (Table 1).

Figures 10b-d show the anomalous winds and the simulated daily mean $O_3$ concentrations at 850 hPa, 950 hPa and the

surface, respectively. The patterns of wind anomalies are similar at these three levels, all of which show a divergence of winds over North China, and anomalous southerlies prevail in this region. The divergence is caused by a high-pressure system at 500 hPa and is represented by index_U500 in the definition of I_OPE. Because $O_3$ concentrations in North China are the highest during OPEs, horizontal advection associated with the divergence has an effect of decreasing $O_3$ concentration in North China, as shown by the IPR analysis.

Currently, among the four indexes that are utilized to define I_OPE, the mechanisms for three of them (index_Tmax, index_RH and index_U500) have been demonstrated. It is of interest to understand the role of index_V850. On the one hand, the anomalous southerlies are associated with the high-pressure system. As Fig. 4b and Fig. 5h show, the strongest southerly anomalies at 850 hPa during OPEs are presented in the west of North China, which is consistent with the southerlies at the west boundary of the anti-cyclone circulation at 500 hPa. On the other hand, the southerlies are likely to have an effect of

increasing the $O_3$ concentrations by transporting $O_3$ during OPEs. Figure 11 presents the composite daily mean $O_3$ concentrations and winds at the surface, 950 hPa and 850 hPa for the first day and the last day of the OPEs. In the composited first day of the OPEs, $O_3$ concentrations in the south of North China are high (Fig. 11a-c). However, when the episodes are ending, $O_3$ concentrations decrease in the south domain but increase in North China (Fig. 11d-f), indicating that the $O_3$ transport strengthens OPEs with the southerly winds.

**6 Conclusions**

In this study, we utilized ground-level observations, reanalyzed meteorological data and a 3-D global transport and chemical model (GEOS-Chem) to understand the ozone pollution events (OPEs) over May-July of 2014-2017 in North China and their relationships with the weather pattern. $O_3$ polluted days in North China are defined as days with an average MDA8 $O_3$ concentration exceeding 160 $\mu g \ m^{-3}$, and OPEs are defined as episodes where $O_3$ pollution lasts for three days or longer.

Ground-based observations showed that North China had the worst $O_3$ pollution in China. There were 167 $O_3$ polluted days and 27 OPEs in North China in the years of 2014-2017, in which 123 $O_3$ polluted days and 21 OPEs occurred in May-July. The mean MDA8 $O_3$ concentrations for OPEs in May to July were 193.0 $\mu g \ m^{-3}$.



A typical weather pattern was identified for OPEs in North China in May–July, which is characterized by high Tmax and low RH at the surface, anomalous southerlies and divergence in the lower troposphere, an anomalous high-pressure system at 500 hPa, and downward air flow from 500 hPa to the surface. The hot and dry air accelerates chemical production of $O_3$ in the boundary layer. The anomalous downward air flow under the high-pressure system transports $O_3$ formed in the upper

layers to the boundary layer. The anomalous southerlies associated with the high-pressure system transport $O_3$ from the south to North China, enhancing the intensity of OPEs. Four parameters, including Tmax, RH, V850 and U500, were selected to define a standardized index I_OPE to represent such a weather pattern. In May-July of 2014-2017, 82 % (101/123) of $O_3$ polluted days and 81 % (17/21) of OPEs occurred with I_OPE>0, indicating that I_OPE has the potential to be used for forecasting OPEs in North China

Integrated process rate (IPR) analysis was applied in the GEOS-Chem model to quantify the contributions of each process (including net chemical production, diffusion, dry deposition, horizontal advection and vertical advection) to OPEs in North China. Relative to the mean condition, net chemical production, diffusion, dry deposition, horizontal advection, and vertical advection during OPEs change by 3.3, -1.2, -0.4, -11.4, and 10.4 Gg $O_3$ day$^{-1}$, indicating that net chemical production, horizontal advection, and vertical advection are the most dominant processes that lead to OPEs. In North China, during

OPEs, net chemical production has a high value at altitudes of 900 to 800 hPa and $O_3$ generated is transported downward to increase $O_3$ at the surface, whereas horizontal advection reduces surface $O_3$.

**Data availability**

The observed hourly ozone concentrations are derived from the Data Center of China's Ministry of Ecology and Environment (http://datacenter.mep.gov.cn/websjzx/queryIndex.vm) over 2014-2017. The NCEP reanalyzed dataset is

obtained from https://www.esrl.noaa.gov/psd/cgi-bin/data/getpage.pl. The GEOS-Chem model is an open-access model managed by the Atmospheric Chemistry Modeling group at Harvard University with support from institutes in North America, Europe, and Asia. The source codes, as well as the MERRA2 reanalyzed data, can be downloaded from http://acmg.seas.harvard.edu/geos/.

**Author contribution**

HL and CG conceived the study and designed the experiments. CG carried out the simulations and performed the analysis. CG and HL prepared the manuscript with contributions from all coauthors.

**Competing interests**

The authors declare that they have no conflict of interest.



**Acknowledgements**

This work was supported by the National Natural Science Foundation of China under grants 91544219, 91744311, and 41475137. We acknowledge the Data Center of China's Ministry of Ecology and Environment and NCAR teams for making their data publicly available. We acknowledge the efforts of the GEOS-Chem working groups for developing and managing the model.

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



**Tables**

**Table 1: Mass flux (MF (Gg O$_3$ day$^{-1}$)) and percentage contributions (PC (%)) of different processes to O$_3$ in North China (36.5°N-40.5°N, 114.5°E-119.5°E) from surface to 850 hPa.**

| | Average [a] | | OPEs [b] | | OPEs-Average [c] |
|---|---|---|---|---|---|
| | MF (Gg O$_3$ day$^{-1}$) | PC (%) | MF (Gg O$_3$ day$^{-1}$) | PC (%) | MF (Gg O$_3$ day$^{-1}$) |
| Net Chemical production | 9.6 | 41.2 | 12.9 | 39.9 | +3.3 |
| Diffusion | 0.4 | 1.6 | -0.8 | -2.5 | -1.2 |
| Dry deposition | -4.9 | -20.9 | -5.3 | -16.5 | -0.4 |
| Horizontal advection | 1.7 | 7.2 | -9.7 | -30.0 | -11.4 |
| Vertical advection | -6.8 | -29.1 | 3.6 | 11.1 | +10.4 |

[a]Average indicates the mean MF and PC averaged over May to July of 2014-2017. [b]OPEs indicate the averaged MF and PC for the 14 OPEs that are captured by the GEOS-Chem model with I_OPE>0. [c]OPEs-Average indicate the differences in MF between OPEs and Average.



**Figures**

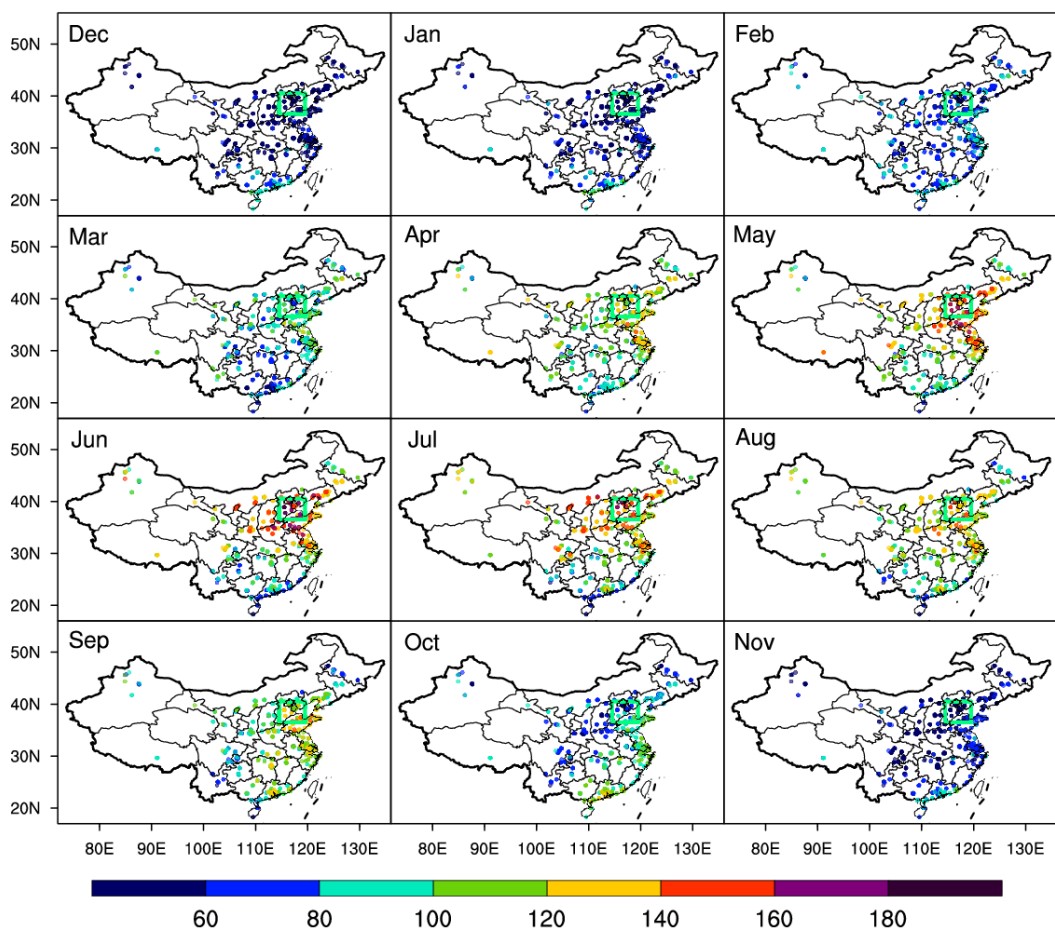

**Figure 1: Monthly MDA8 O$_3$ concentrations (μg m$^{-3}$) averaged over 2014-2017 at 740 observational sites. The green solid lines enclose the North China region.**


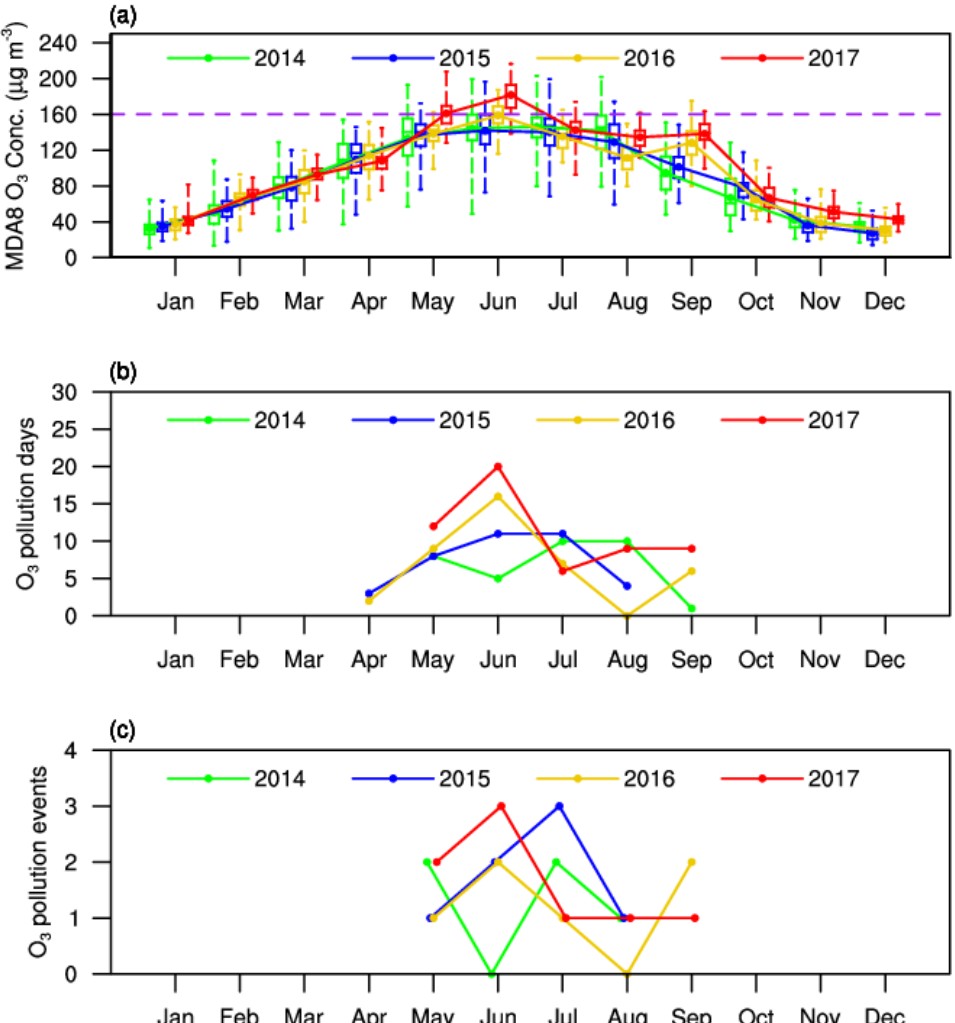

**Figure 2: (a) Monthly variation in MDA8 O₃ concentration (μg m⁻³) averaged over North China for 2014 to 2017. The boxes indicate the maximum and minimum MDA8 O₃ concentrations for 62 observational sites in North China. Dotted solid lines denote the averaged values in North China. The purple dashed line indicates the threshold of 160 μg m⁻³ for O₃ polluted days. (b) Monthly variation of O₃ polluted days in North China for 2014-2017. (c) Monthly variation of the number of ozone polluted events (OPEs) in North China for 2014-2017.**





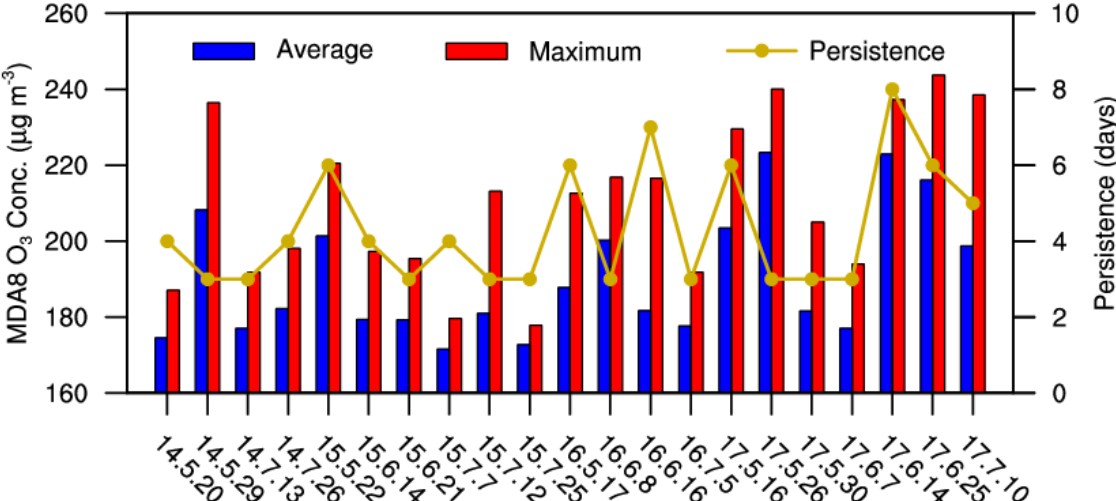

**Figure 3: Mean (blue bars) and maximum (red bars) MDA8 O$_3$ concentrations (µg m$^{-3}$) averaged over North China for each of the 21 OPEs that occurred during May-July of 2014-2017. The dotted yellow line indicates the persistence (days) of each OPE.**





**Figure 4: Composites of (a) wind field and geopotential height at 500 hPa, (b) wind field and geopotential height at 850 hPa, (c) surface wind field and SLP, (d) Tmax at the surface, (e) RH at the surface, (f) pressure-latitude cross-section of vertical pressure velocity (ω, positive value indicates downward air flow), and (g) pressure-latitude cross-section of divergence (positive value indicates divergence) for the 21 OPEs in North China. The data shown are composited over the detrended and standardized time series during May-July of 2014-2017 (see Sect. 4.1). The green solid lines enclose North China. The red vectors in (a)-(c) highlight the important circulation features for OPEs. The cross-sections are averaged over the longitudes of 114.5°E-119.5°E.**



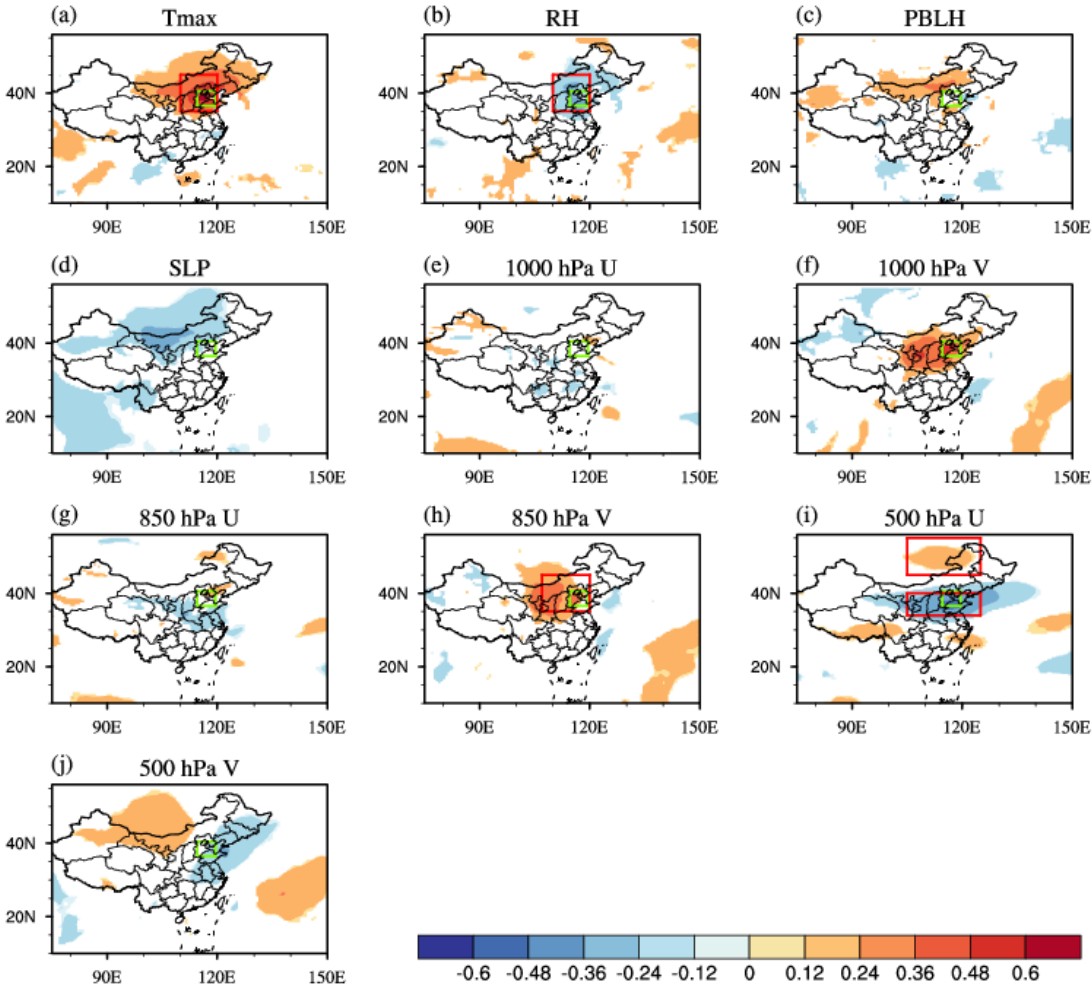

**Figure 5:** Correlation coefficients, for May to July of 2014-2017, between daily regional mean MDA8 O₃ concentrations in North China and daily mean (a) 2-meter Tmax at the surface, (b) RH at the surface, (c) planetary boundary layer height, (d) surface level pressure, (e) meridional winds at 1000 hPa, (f) zonal winds at 1000 hPa, (g) meridional winds at 850 hPa, (h) zonal winds at 850 hPa, (i) meridional winds at 500 hPa, and (j) zonal winds at 500 hPa in Asia. Colored regions are correlation coefficients that are statistically significant above the 99 % confidence level. The red rectangles in (a), (b), (h) and (i) denote the regions for calculating index_Tmax, index_RH, index_V850 and index_U500, respectively (see Sect. 4.3). The green rectangle indicates the region of North China.





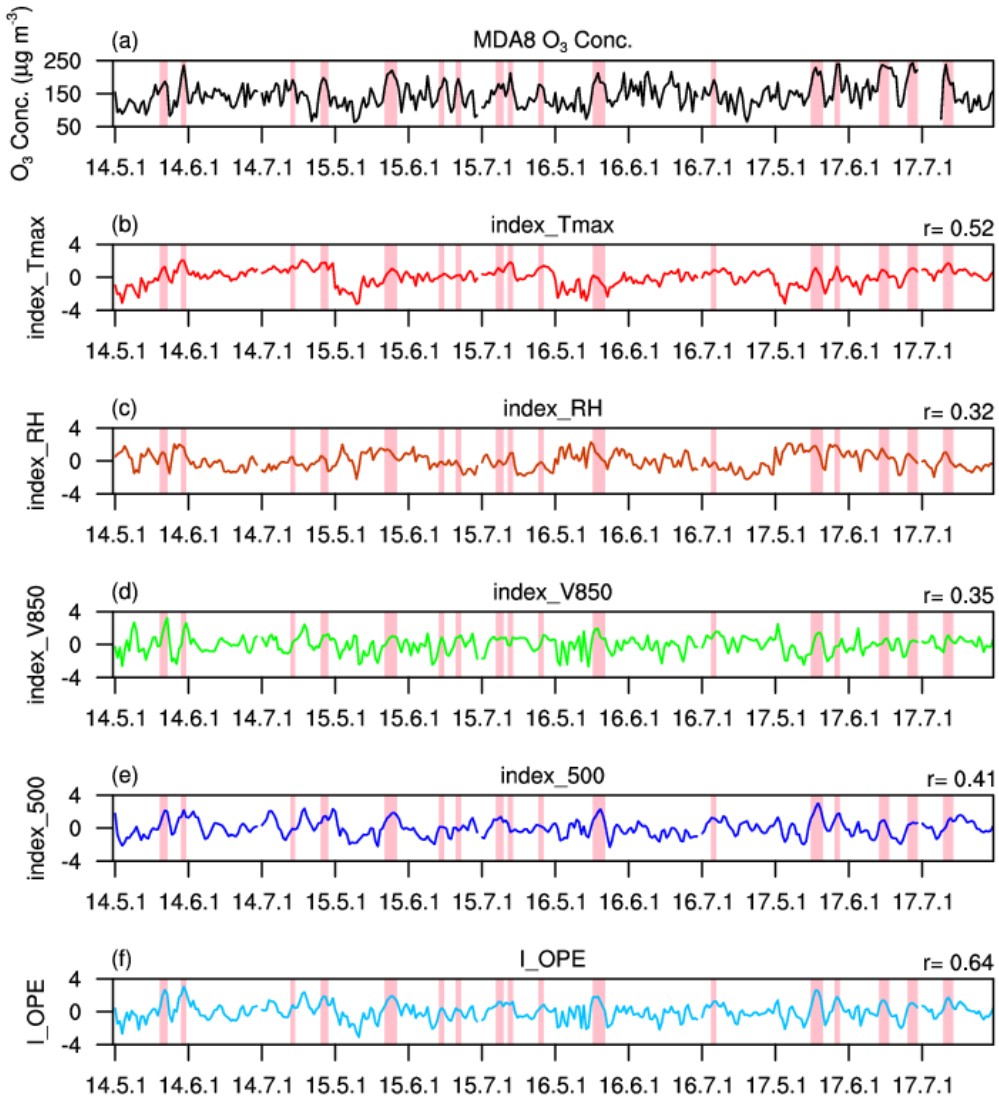

**Figure 6: Daily variations in (a) MDA8 O$_3$ concentrations (µg m$^{-3}$) in North China, (b) index_Tmax, (c) index_RH, (d) index_V850, (e) index_U500, and (f) I_OPE for May-July of 2014-2017. Observed OPEs in North China are highlighted by pink rectangles. Correlation coefficients between MDA8 O$_3$ concentrations and different indexes are shown above the top right corner of each plot.**

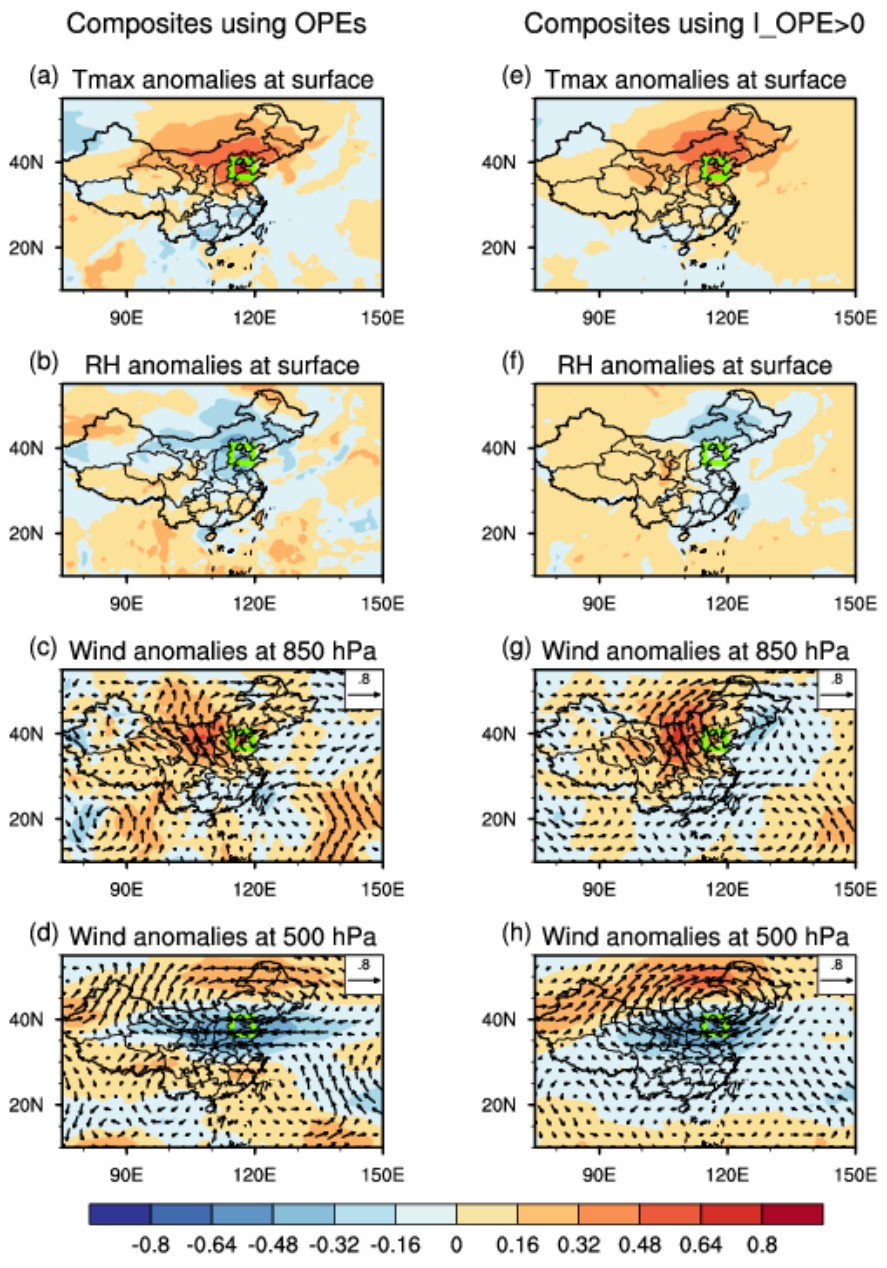

**Figure 7: Weather conditions for OPEs in North China.** Left column shows composites of weather conditions for observed OPEs (on the basis of observed O₃ concentrations) for (a) anomalous Tmax, (b) anomalous RH, (c) anomalous wind vectors at 850 hPa (shades indicate meridional flow) and (d) anomalous wind vectors at 500 hPa (shades indicate zonal flow). (e)-(h), the same as (a)-(d), respectively, but show composites of weather conditions for days with I_OPE>0. The data shown are composited over the detrended and standardized time series during May-July of 2014-2017 (see Sect. 4.1). The green solid lines enclose North China

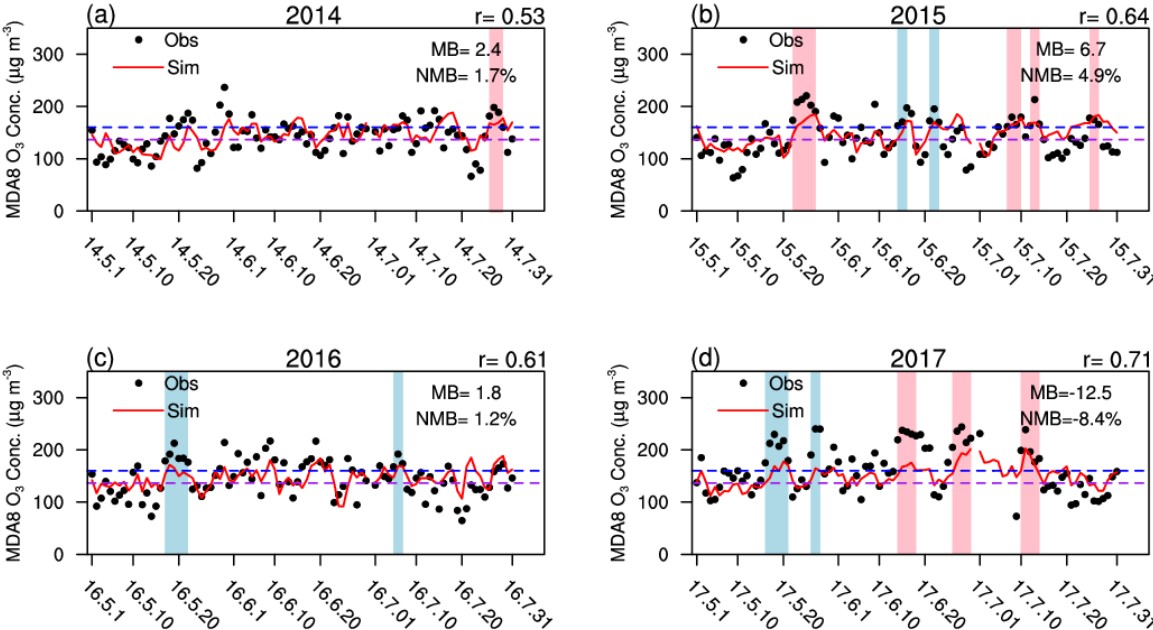

**Figure 8: Daily variations in observed (black dots) and simulated (red solid lines) regional mean MDA8 O$_3$ (μg m$^{-3}$) in North China during May to July of 2014-2017. The blue and purple dashed lines indicate the thresholds of 160 μg m$^{-3}$ and 136.6 μg m$^{-3}$ for observation and simulation, respectively. OPEs captured by the GEOS-Chem model with I_OPE>0 are highlighted by pink rectangles (OPEs with simulated MDA8 O$_3$ concentrations larger than 160 μg m$^{-3}$) and by blue rectangles (OPEs with simulated MDA8 O$_3$ concentrations larger than 136.6 μg m$^{-3}$ but including days with simulated MDA8 O$_3$ smaller or equal to 160 μg m$^{-3}$). Correlation coefficient between observed and simulated MDA8 O$_3$ concentrations for each year is shown above the top right corner of each plot. The mean bias (MB) and normalized mean bias (NMB) are calculated by $MB = \frac{1}{n}\sum_i^n(S_i - O_i)$ and $NMB = \sum_i^n(S_i - O_i) / \sum_i^n O_i * 100$ %, where $O_i$ and $S_i$ indicate the observed and simulated MDA8 O$_3$ concentrations on the $i$ day, respectively, and $n$ indicates the number of days.**



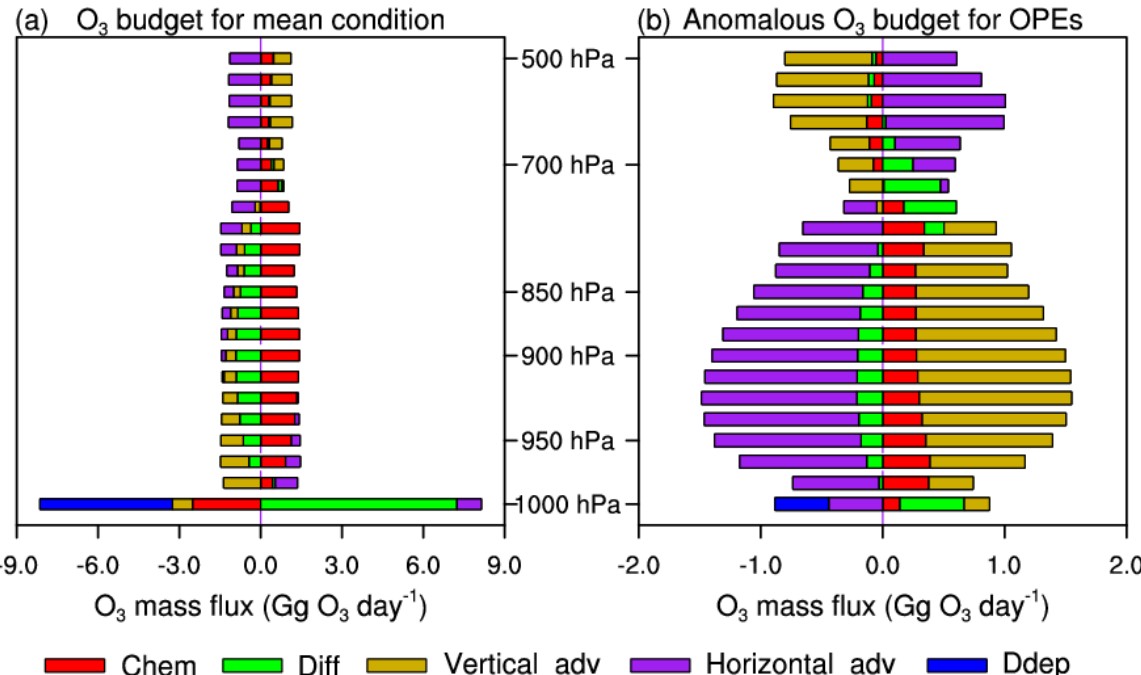

**Figure 9: (a) Vertical profile of O₃ mass flux (Gg O₃ day⁻¹) over North China for each process that is averaged over all days in May-July of 2014-2017. (b) Anomalous vertical profile of each process during the 14 OPEs relative to the mean value of May-July of 2014-2017. The 14 OPEs are captured by the GEOS-Chem model with I_OPE>0 in North China.**


**Figure 10:** (a) The pressure-latitude cross-section averaged over the longitudes of 114.5 °E-119.5 °E of simulated daily mean O$_3$ concentrations (µg m$^{-3}$) during the 14 OPEs that are captured by the GEOS-Chem model with I_OPE>0. The red line with asterisks shows the anomalous profile of the regionally averaged vertical pressure velocity (ω, Pa s$^{-1}$, positive value indicates downward airflow) in North China. The purple dashed line indicates the position where the standardized ω is zero. (b)-(d) show anomalous winds and the simulated daily mean O$_3$ concentrations during OPEs at (b) 850 hPa, (c) 950 hPa and (d) the surface. The green solid lines enclose North China. ω in (a) and winds in (b)-(d) are composited over the detrended and standardized time series during May-July of 2014-2017 (see Sect. 4.1)





**Figure 11: Winds (m s⁻¹) and the simulated O₃ concentrations (µg m⁻³) averaged over the first day of the 14 OPEs that are captured by the GEOS-Chem model with I_OPE>0 at (a) 850 hPa, (b) 950 hPa and (c) the surface. (d)-(f) are the same as (a)-(c) but are averaged over the final day of the OPEs. The green solid lines enclose North China.**