# Peer review of "A typical weather pattern for the ozone pollution events in North China"

_Atmospheric Chemistry and Physics, 2019_

## Referee Comment (RC1) · Anonymous Referee #1 · 18 Jul 2019

This study examined the possible mechanisms for the ozone pollution events (OPEs) in North China during 2014-2017 using GEOS-Chem model together with an integrated process rate (IPR) analysis. They found that OPEs in North China occurred under a typical weather pattern with high daily maximum temperature, low relative humidity, anomalous southerlies and an anomalous downward air flow caused by an anomalous high-pressure system at 500 hPa. The topic is of interest, the method is sound. I would suggest for publication after addressing my comments below.

Page 2 Lines 11-13: Please reframe this sentence.

Page 4 Lines 14-15: I don't think the original resolution of MERRA2 data is the same as GEOS-Chem model. The meteorological data authors used are modified to fit the model resolution.

Page 4 Lines 23-24: How did the authors detrend the meteorological parameters to remove interannual or seasonal variability? Please specify the method or provide formula they used.

Page 5 Line 7: The annual emission from 2014 to 2017 are applied in the simulation, but the authors did not rule out the impacts of changing emissions on the OPEs selection and IPR analysis, although the changes in emissions in the four years are not likely to be very large.

Figures: All the figures and analysis are lack of significance test. Please add in.

Page 7 Line 9: I on day 'd'.

Page 8 Line 9: It should be 850 hPa 'meridional winds' and 500 hPa 'zonal' winds.

Page 11 Line 9: Before analyzing vertical profiles of each process, the authors should give vertical profile of O3 concentrations in terms of seasonal mean and anomalies during OPEs.

Page 11 Line 24: 'horizontal advection' is the compensating from the increasing ozone from the figure. I don't think it should be listed as the dominant processes that lead to OPEs, although the negative value is large.

---

## Referee Comment (RC2) · Anonymous Referee #2 · 22 Jul 2019

General comments: Ozone pollution in China is becoming a noticeable problem particularly in summer season. This paper focuses on this problem in north China region. Two parts of work have done. One is a long term (4 years) analysis of the ozone pollution status. Ozone pollution days and events are defined and identified in the research years. Using these days/events, the so called correspondent weather pattern are composited. The second part of work is to establish an index to identify the ozone pollution day/event. Using GEOS-Chem model, simulation results for these 4 summers are used to support the index.

The ozone pollution status is clearly shown. The related weather pattern seems a reasonable but anticipative result. The GEOS-Chem simulation provides results not so informative.

[Figure]

Specific comments: 1)It is well known that the ozone pollution is related to sunny days, high temperature, precursors, boundary layer process, etc. Once the high ozone events are selected, the statistics of weather pattern is just a conditional sampling result, so that the features are anticipative.

2)The selection of Index_U500 seems quite arbitrary. What does it mean by the wind speed difference of two zones? What is the reason to choose these two zones? Is it ok the zones larger or smaller?

3)GEOS-Chem simulation of ozone concentration does not agree to the observation satisfactorily in Figure 8.

4)The role of diffusion or mixing on ozone mass flux is not clearly described. At first, the authors declare "Note that the MF of each process at a specific level indicates the net O3 mass change within this level rather than the flux across this level, especially for the vertical processes such as diffusion and vertical advection", but at later, they state "it is indicated that O3 aloft is transported downward to be mixed at the surface by the diffusion process", and " Vertical advection exhibits negative MF values from the surface to approximately 750 hPa". We need to clarify it is the "mass flux" or the "mass flux divergence", the former indicates mass across the level, the latter is the net mass change.

5)I think the ozone production is mainly within the atmospheric boundary layer, not above it. So it is not true: "hot air is beneficial for developing the mixed layer, leading to more O3 mixed downward to the surface during OPEs".

Other points:

0) too many acronym, someone not necessary, for example, mass flux: MF. 1)Page 1 line19: "chemical production of O3 was high between 800 and 900 hPa", what height? 2)Page 3, line 22: "Section 3 presents the observed and spatiotemporal distributions of OPEs in North China during 2014 to 2017", sentence not very clear. 3)page 3:
(http://datacenter.mep.gov.cn/websjzx/queryIndex.vm), no linkage. 4)page 4, line 2: "(67 sites among the 114 sites in North China (36ïĊř-40.5ïĊřN, 114.5ïĊř-119.5ïĊřE)) are selected and used in this study." Need more details or figure to show the 67 sites. 5)Page 4, line 20: "MERRA2 dataset, daily mean geopotential heights at 850 hPa and 500 hPa from the National Center for Environmental Prediction (NCEP)...", MERRA2 and NCEP dataset, consistent? At least the resolution is different. 6)page 4, line 24: "All the time series of meteorological parameters have been detrended first and then standardized by their respective standard deviation to remove interannual or seasonal variability", what is the performance and result? 7)page 6 line 1: "all 62 sites", previously 67 sites! 8)page 6, line 24: "last for many consecutive days. The mean duration of OPEs is 4.3 days, while some episodes can last for one week and even Longer". Can be interpreted by sub-tropical high, in summer. 9)page 8 line 10: "850 hPa zonal winds indicate circulation in the lower atmosphere and 500 hPa meridional winds describe the dominate large-scale circulation", why take zonal winds at 850hPa? 10)Figure 5: why the calculation region for V_850hPa differently? 11)Figure 7, similar to Figure 4? 12)Page 9 line 21: "indicating that the GEOS-Chem model can simulate the daily variation in MDA8 O3", but the simulation not agree well to the observation in Figure 8. 13)page 9 line 26: " The linear regression through the origin between observed and simulated MDA8 O3 has a regression coefficient of 0.96, indicating the capability of the model in simulating the MDA8 O3 concentrations." Need to be clarified. 14)page 10, line 29: "diffusion has a small effect on the total mass of O3 in the boundary layer. However, the diffusion process is important in 30 the boundary layer, which has...", confused. 15)page 10 line 31: "mass flux of diffusion for the whole boundary layer is small, it is indicated that O3 aloft is transported downward to be mixed at the surface by the diffusion process", very strange explanation. 16)Page 11 line 13: "both the positive contribution at the surface and the negative contributions in the upper layers increase", ok. "indicating that more O3 is mixed from the upper levels to the surface to increase the surface O3 concentration during OPEs", why mixing/diffusion increase surface O3? 17)Page 12 line 1: "beneficial for developing the mixed layer, leading to more O3 mixed

downward to the surface during OPEs", O3 produces in the boundary layer, no need to mixing down from upper layer above ABL. 18)page 13 line 14: "horizontal advection, and vertical advection are the most dominant processes that lead to OPEs", but horizontal advection contributes negative mass flux? So, not lead to OPEs, but depress the development of OPEs.

---

## Author Comment (AC1) · 18 Sep 2019

**Response to Comments of Reviewer #1**

**Manuscript number:** acp-2019-263

**Authors:** Cheng Gong and Hong Liao

**Title:** A typical weather pattern for the ozone pollution events in North China

**General comments:**

*This study examined the possible mechanisms for the ozone pollution events (OPEs) in North China during 2014-2017 using GEOS-Chem model together with an integrated process rate (IPR) analysis. They found that OPEs in North China occurred under a typical weather pattern with high daily maximum temperature, low relative humidity, anomalous southerlies and an anomalous downward air flow caused by an anomalous high-pressure system at 500 hPa. The topic is of interest, the method is sound. I would suggest for publication after addressing my comments below.*

**Response:**

Thanks to the reviewer for the helpful comments and suggestions. We have revised the manuscript carefully and the point-to-point responses are listed below.

**Specific Comments:**

*Page 2 Lines 11 -13: Please reframe this sentence*

**Response:**

We have reframed this sentence as:

'Zhang et al. (2015) showed that values of RH for days with top 10% $O_3$ concentrations were lower compared to those for days with bottom 10% $O_3$ concentrations by examining continuous observations of $O_3$ and meteorological parameters in Guangzhou during March 2013 to February 2014.'

*Page 4 Lines 14-15: I don't think the original resolution of MERRA2 data is the same as GEOS-Chem model. The meteorological data authors used are modified to fit the model resolution.*

**Response:**

We have revised the sentence to clarify: 'The original MERRA2 data has a horizontal resolution of 0.5° latitude x 0.625° longitude and 72 vertical layers (Molod et al., 2015). The GEOS-Chem model has the same horizontal resolution over the nested domain but the GEOS-Chem support team has lumped the 72 vertical layers into 47 layers to save computational resources. The lumped vertical levels are within the 32[th] model layer (about 190 hPa) and the top of atmosphere (about 0.01 hPa).'

*Page 4 Lines 23-24: How did the authors detrend the meteorological parameters to*

*remove interannual or seasonal variability? Please specify the method or provide formula they used.*

**Response:**

Following the reviewer's comments, we have compared our analyses with and without detrending and found small impact on our results because of the relatively short time period (only 4 years over 2014-2017). To avoid confusion, we have removed the detrending process and updated the table and figures in the revised manuscript. The description here has been revised as follows:

'The daily time series of a meteorological parameter $x$ at a specific model grid cell over May to July of 2014-2017 is standardized by:

$$[x_i] = \frac{x_i - \frac{\Sigma_i^n x_i}{n}}{s_i} \tag{1}$$

where $x_i$ indicates the parameter $x$ on day $i$, $n$ is the total number of days over May to July in 2014-2017, $s_i$ indicates the standard deviation of the daily time series. $[x_i]$ is the standardized anomaly for parameter $x$ on day $i$.'

*Page 5 Line 7: The annual emission from 2014 to 2017 are applied in the simulation, but the authors did not rule out the impacts of changing emissions on the OPEs selection and IPR analysis, although the changes in emissions in the four years are not likely to be very large.*

**Response:**

We use emissions from 2014 to 2017 in the model to obtain OPEs with realistic changes in emissions. Following your suggestion, we have carried out a new simulation by fixing anthropogenic emissions at year 2014 levels. Twelve of the 17 observed OPEs with I_OPE >0 can be identified by applying the same threshold (136.6 $\mu$g m$^{-3}$) in the model (Figure R1). Compared with the simulation with year-by-year changes in emissions from 2014 to 2017, three OPEs (one in June of 2015, one in July of 2016, and one in May of 2017) are missed in the run with fixed emissions. The results from IPR analysis with fixed emissions are similar to those with changes in emissions except that the simulation with fixed emissions has lower changes in $O_3$ mass by net chemical production due to the changes in $NO_x$/VOCs ratio (Li et al., 2019). As a result, the changes in emissions have little impacts on the OPEs selection and IPR analysis (Figures R1 and R2 and Table R1).

[Figure]

Figure R1. The same as Figure 7 in the revised manuscript but with fixed emissions at 2014 levels.

[Figure]

Figure R2. The same as Figure 8 in the revised manuscript but with fixed emissions at 2014 levels

Table R1. The same as Table 1 in the revised manuscript but with fixed emissions at 2014 levels.

| | Average [a] | | OPEs [b] | | OPEs-Average [c] |
|---|---|---|---|---|---|
| | NC | PC | NC | PC | MF |
| | (Gg O$_3$ day$^{-1}$) | (%) | (Gg O$_3$ day$^{-1}$) | (%) | (Gg O$_3$ day$^{-1}$) |
| Net Chemical production | 4.6 | 21.8 | 7.5 | 28.8 | +2.9 |
| Diffusion | 2.4 | 11.4 | 2.1 | 8.1 | -0.3 |
| Dry deposition | -4.3 | -20.4 | -4.8 | -18.5 | -0.5 |
| Horizontal advection | 3.5 | 16.6 | -8.0 | -30.8 | -11.5 |
| Vertical advection | -6.3 | -29.8 | 3.6 | 13.8 | +9.9 |

*Figures: All the figures and analysis are lack of significance test. Please add in.*

**Response:**

We have added the significance test with 95 % confidence in Figures 4, 5, 9 and S2 in the revised manuscript and supplementary material.

*Page 7 Line 9: I on day 'd'.*

**Response:**

The 'd' has been added.

*Page 8 Line 9: It should be 850 hPa 'meridional winds' and 500 hPa 'zonal' winds.*

**Response:**

Corrected.

*Page 11 Line 9: Before analyzing vertical profiles of each process, the authors should give vertical profile of O3 concentrations in terms of seasonal mean and anomalies during OPEs.*

**Response:**

Following the reviewer's suggestion, we have added a new panel in Fig. 8 (Fig. 8a) in the revised manuscript to show the vertical profiles of $O_3$ concentrations in terms of seasonal mean and anomalies during OPEs. We have also added the following sentences to describe these vertical profiles of $O_3$ in the text:

'The vertical profile of simulated daily $O_3$ concentrations averaged over May to July in 2014-2017 as well as that composited over the 15 OPEs are shown in Fig. 8a. For both profiles, the $O_3$ concentrations are highest between 950 hPa and 850 hPa and are relatively lower at the surface due to the titration by high $NO_x$ concentrations. When OPEs occur, $O_3$ concentrations are higher from the surface to 700 hPa (about 3 km altitude) but change little above 700 hPa, indicating that the enhancement of $O_3$ concentrations during OPEs occurs not only at the surface but also in and above the boundary layer.'

[Figure]

Figure 8. (a) Vertical profile of simulated daily $O_3$ concentrations (μg m$^{-3}$) averaged over May to July in 2014-2017 (blue line and triangle) as well as that composited over the 15 simulated OPEs with I_OPE>0 (red line and triangle) in North China. (b) Vertical profile of $O_3$ mass flux (Gg $O_3$ day$^{-1}$) over North China for each process that is averaged over all days in May-July of 2014-2017. (c) Anomalous vertical profile of each process during the 15 OPEs relative to the mean value of May-July of 2014-2017. The vertical axis is the same for all the panels with a unit of hPa.

*Page 11 Line 24: 'horizontal advection' is the compensating from the increasing ozone from the figure. I don't think it should be listed as the dominant processes that lead to OPEs, although the negative value is large.*

**Response:**

'horizontal advection' has been removed.

**References:**

Li, K., Jacob, D. J., Liao, H., Shen, L., Zhang, Q., and Bates, K. H.: Anthropogenic drivers of 2013-2017 trends in summer surface ozone in China, Proceedings of the National Academy of Sciences of the United States of America, 116, 422-427, 10.1073/pnas.1812168116, 2019.

Molod, A., Takacs, L., Suarez, M., and Bacmeister, J.: Development of the GEOS-5 atmospheric general circulation model: evolution from MERRA to MERRA2, Geoscientific Model Development, 8, 1339-1356, 10.5194/gmd-8-1339-2015, 2015.

---

## Author Comment (AC2) · 18 Sep 2019

**Response to Comments of Reviewer #2**

**Manuscript number:** acp-2019-263

**Authors:** Cheng Gong and Hong Liao

**Title:** A typical weather pattern for the ozone pollution events in North China

**General comments:**

*General comments: Ozone pollution in China is becoming a noticeable problem particularly in summer season. This paper focuses on this problem in north China region. Two parts of work have done. One is a long term (4 years) analysis of the ozone pollution status. Ozone pollution days and events are defined and identified in the research years. Using these days/events, the so called correspondent weather pattern are composited. The second part of work is to establish an index to identify the ozone pollution day/event. Using GEOS-Chem model, simulation results for these 4 summers are used to support the index.*

*The ozone pollution status is clearly shown. The related weather pattern seems a reasonable but anticipative result. The GEOS-Chem simulation provides results not so informative*

**Response:**

Understanding the weather pattern that leads to OPEs is important for better understanding the formation of OPEs and for forecasting OPEs on daily scale. Previous studies that examined OPEs and the associated weather patterns in China were generally focused on one or two episodes of high $O_3$ concentrations at specific locations, such as Mountains Tai and Huang (Wang et al., 2006), Hangzhou (Li et al., 2017a), Shanghai and Nanjing (Shu et al., 2016). Our work reports a typical 3-D weather pattern for OPEs in North China on the basis of national air quality monitoring data and reanalyzed meteorological fields for 2014-2017, which is a more representative and systematic investigation compared with previous studies.

The typical weather pattern is characterized by high temperature and low humidity at the surface, anomalous southerlies and divergence in the lower troposphere (from surface to 850 hPa), high pressure system at 500 hPa and downward air flows from 500 hPa to the surface. Although high temperature and low humidity have been reported in previous studies (e.g. Zhang and Wang, 2016; Pu et al., 2017; Zhang et al., 2017), we find some new features for the formation of OPEs in North China (such as the downward airflow and southerlies).

We carry out process analysis using the GEOS-Chem model to identify the dominant processes that lead to OPEs, which, to our knowledge, is the first study to have such quantitative examination of the weather pattern to understand the mechanisms for the formation of OPEs. Our analyses show that the net chemical production is the most dominate process for the seasonal mean condition, however, when OPEs occur, the most dominant process is vertical advection that leads to the largest net increase in $O_3$

mass from the surface to 850 hPa. We have added a schematic diagram of the typical weather pattern showing the mechanisms for the formation of OPEs in North China (a new Fig. 11 in the revised manuscript).

[Figure]

Figure 11. A schematic diagram of the typical weather pattern showing the mechanisms for the formation of OPEs in North China

**Specific Comments:**

1. *It is well known that the ozone pollution is related to sunny days, high temperature, precursors, boundary layer process, etc. Once the high ozone events are selected, the statistics of weather pattern is just a conditional sampling result, so that the features are anticipative.*

**Response:**

See our responses to your general comments.

2. *The selection of Index_U500 seems quite arbitrary. What does it mean by the wind speed difference of two zones? What is the reason to choose these two zones? Is it ok the zones larger or smaller?*

**Response:**

The main purpose of using index_U500 is to represent the high-pressure system at 500 hPa level during OPEs relative to the seasonal mean conditions (Fig. 4a). Since the high-pressure system is characterized by anti-cyclone circulation, the index_U500 is defined as the difference in zonal winds (westerlies are positive) between the northern region (supposed to be westerlies) and the southern region (supposed to be easterlies) of the typical high-pressure system (Eq. (7)). As a result, the index_U500 can be used to describe whether the high-pressure system exists (index_U500 >0) or not (index_U500<0). Higher index_U500 indicates stronger anti-cyclone circulation

and stronger high-pressure system. A similar method has been used in the previous study of Cai et al. (2017).

The regions for the calculation of index_x (including index_U500) in Eq.3 are selected on the basis of the correlations between MDA8 $O_3$ concentrations in North China and the corresponding meteorological parameters (Fig. 5). Figure 5i shows that, for correlations between MDA8 $O_3$ concentrations and the zonal winds at 500 hPa, the correlation coefficients are the largest in the two regions enclosed by red rectangles; therefore these two regions are used for the definition of index_U500.

3. *GEOS-Chem simulation of ozone concentration does not agree to the observation satisfactorily in Figure 8.*

**Response:**

The GEOS-Chem model has been used to simulate $O_3$ in China and been evaluated extensively in previous studies (Wang et al., 2011; Yang et al., 2014; Lou et al., 2014; Lou et al., 2015; Ni et al., 2018; Li et al., 2019; Lu et al., 2019; Sun et al., 2019), which shows the GEOS-Chem model can capture fairly well the daily, monthly, seasonal, and interannual variations of $O_3$ in China. In our work, we evaluate mean bias (MB) and normalized mean bias (NMB) of simulated MDA8 $O_3$ concentrations averaged over North China by comparing with measurements. For the daily time series of MDA8 $O_3$ concentrations over May to July in 2014-2017, simulated concentrations have a mean MB (NMB) of 2.4 μg m$^{-3}$ (1.7%) in 2014, 6.7 μg m$^{-3}$ (4.9%) in 2015, 1.8 μg m$^{-3}$ (1.2%) in 2016, and -12.5 μg m$^{-3}$ (-8.4%) in 2017 (Figure 7), indicating that the GEOS-Chem model has a good performance. We do find that the GEOS-Chem model has difficulties in capturing the peak values of $O_3$ concentrations, which is a common issue in the GEOS-Chem model (Zhang and Wang, 2016; Ni et al., 2018), WRF-Chem (Tie et al., 2009) and WRF-CMAQ (Shu et al., 2016). In our analysis, the threshold for OPEs in the model has been revised as 136.6 μg m$^{-3}$ (160*85.4 %) by applying the NMB of -14.6 % for the days with observed MDA8 $O_3$ > 160 μg m$^{-3}$. This modification enables us to identify 15 of 21 observed OPEs with I_OPE>0.

4. *The role of diffusion or mixing on ozone mass flux is not clearly described. At first, the authors declare "Note that the MF of each process at a specific level indicates the net O3 mass change within this level rather than the flux across this level, especially for the vertical processes such as diffusion and vertical advection", but at later, they state "it is indicated that O3 aloft is transported downward to be mixed at the surface by the diffusion process", and " Vertical advection exhibits negative MF values from the surface to approximately 750 hPa". We need to clarify it is the "mass flux" or the "mass flux divergence", the former indicates mass across the level, the latter is the net mass change.*

**Response:**

Thanks for the comments. To avoid confusion and also to take into account your comment on too many acronym (Other point #0), we have replaced 'mass flux' or MF

in the text by 'net change in $O_3$ mass' when we describe IPR for each process in a specific model layer.

5. *I think the ozone production is mainly within the atmospheric boundary layer, not above it. So it is not true: "hot air is beneficial for developing the mixed layer, leading to more O3 mixed downward to the surface during OPEs"*

**Response:**

As shown in Figure 8, $O_3$ production is large not only within the boundary layer (from 850 hPa to the surface) but also between 850 and 800 hPa, especially during the OPEs. We highlight that the vertical concentration gradient caused by $O_3$ chemical production at and above the upper boundary layer and chemical loss at the surface leads to downward transport of $O_3$ by diffusion process. We have revised this sentence to clarify:

'Moreover, hot and sunny weather during OPEs increases the vertical concentration gradient (stronger chemical production at and above the upper boundary layer), leading to more $O_3$ transported downward to the surface as described in Sect. 5.2.2.'

**Other points:**

0. *too many acronym, someone not necessary, for example, mass flux: MF.*

**Response:**

We have replaced 'mass flux' or MF in the text by 'net change in $O_3$ mass' when we describe IPR for each process in a specific model layer.

1. *Page 1 line19: "chemical production of O3 was high between 800 and 900 hPa", what height?*

**Response:**

The GEOS-Chem model describes vertical layers by hPa (see *http://wiki.seas.harvard.edu/geos-chem/index.php/GEOS-Chem_vertical_grids*). We have clarified here 'chemical production of $O_3$ was high between 800 and 900 hPa (approximately 0.8-1.8 km altitudes)'.

2. *Page 3, line 22: "Section 3 presents the observed and spatiotemporal distributions of OPEs in North China during 2014 to 2017", sentence not very clear.*

**Response:**

This sentence has been revised as:

'Section 3 presents the observed frequency and intensity of OPEs in North China during 2014-2017.'

3. *page 3: (http://datacenter.mep.gov.cn/websjzx/queryIndex.vm), no linkage.*

**Response:**

Since the name of Ministry of Environmental Protection (MEP) was changed to Ministry of Ecology and Environment (MEE), the website address is now http://datacenter.mee.gov.cn/websjzx/queryIndex.vm.

4. *page 4, line 2:"(67 sites among the 114 sites in North China (36°-40.5°N, 114.5°-119.5°E)) are selected and used in this study." Need more details or figure to show the 67 sites.*

**Response:**

Sorry for the inconsistent border over North China here, which should be (36.5°-40.5°N, 114.5°-119.5°E). We have added a new figure (Fig. S1) in the supplementary material to show these 62 sites. The sentence in the text has been revised as: 'As a result, 740 among the 1582 sites in China (62 sites among the 101 sites in North China (36.5°-40.5°N, 114.5°-119.5°E), Fig. S1) are selected and used in this study.'

[Figure]

Figure S1. Distribution of the observational sites in North China. The gray dots indicate sites eliminated by the data quality control (see Sect. 2.1 for details). The red and blue dots indicate the selected sites inside and outside North China, respectively. The green rectangle encloses North China.

5. *Page 4, line 20: "MERRA2 dataset, daily mean geopotential heights at 850 hPa and 500 hPa from the National Center for Environmental Prediction (NCEP) . . .", MERRA2 and NCEP dataset, consistent? At least the resolution is different.*

**Response:**

The NCEP dataset is only used in Figure 4 for geopotential heights due to the lack of geopotential heights in MERRA2 dataset. In Figure 4, all of the meteorological parameters from MERRA2 and NCEP dataset have the same time period (May to July

over 2014-2017), time resolution (daily). The only difference between MERRA2 and NCEP datasets is the different spatial resolution (0.5° latitude x 0.625° longitude for MERRA2 and 2.5° latitude x 2.5° longitude for NCEP). However, it is not a problem in Figure 4 because the drawing software (NCAR Command Language, NCL) we utilized is able to contour the map automatically according to the resolution of the dataset.

6. *page 4, line 24:"All the time series of meteorological parameters have been detrended first and then standardized by their respective standard deviation to remove interannual or seasonal variability", what is the performance and result?*

**Response:**

Following the other reviewer's comments, we have compared our analyses with and without detrending and found small impact on our results, because of the relatively short time period (only 4 years over 2014-2017). To avoid confusion, we have removed the detrending process and updated the table and figures in the revised manuscript. The description here has been revised as follows:

'The daily time series of a meteorological parameter *x* at a specific model grid cell over May to July of 2014-2017 is standardized by:

$$[x_i] = \frac{x_i - \frac{\sum_i^n x_i}{n}}{s_i} \tag{1}$$

where $x_i$ indicates the parameter *x* on day *i*, *n* is the total number of days over May to July in 2014-2017, $s_i$ indicates the standard deviation of the daily time series. $[x_i]$ is the standardized anomaly for parameter *x* on day *i*.'

7. *page 6 line 1: "all 62 sites", previously 67 sites!*

**Response:**

The previous '67 sites' has been revised as '62 sites'.

8. *page 6, line 24: "last for many consecutive days. The mean duration of OPEs is 4.3 days, while some episodes can last for one week and even Longer". Can be interpreted by sub-tropical high, in summer.*

**Response:**

Climatically, the onset of sub-tropical high occurs in central and southern Indochina Peninsula in early May. Then sub-tropical high migrates northward in a stepwise fashion, characterized by two northward jumps in mid-June (to 20°-25°N) and in late July (to 25°-30°N or even north ) (Ding and Chan, 2005; Su et al., 2014). As a result, the sub-tropical high can barely influence North China during our studied time period of May to July.

In synoptic meteorology, the regions with geopotential height larger than 5880 m at

500 hPa level are considered being controlled by sub-tropical high. By applying this definition and the NCEP dataset, the locations of sub-tropical high for each OPE in our analysis are represented (Figure R1). None of the OPEs in North China occurs under the sub-tropical high. Also, by comparing the geopotential height at 500 hPa averaged over May to July in 2014-2017 and the 21 OPEs, we find that the location of sub-tropical high change little (Figure R2, sub-tropical high is highlighted by the black dots). We believe that the high pressure system identified in our study is irrelevant with the sub-tropical high.

[Figure]

Figure R1. The mean geopotential height (m) at 500 hPa level for each OPE in North China. Only regions with geopotential height larger than 5880 m are colored to represent the locations of the sub-tropical high.

[Figure]

Figure R2. The geopotential height (m) at 500 hPa averaged over May to July in 2014-2017 (left) and over the 21 observed OPEs (right). The locations of sub-tropical high (geopotential height larger than 5880 m) are highlighted by the black dots.

9. *page 8 line 10: "850 hPa zonal winds indicate circulation in the lower atmosphere and 500 hPa meridional winds describe the dominate large-scale circulation", why take zonal winds at 850hPa?*

**Response:**

It has been revised as '850 hPa meridional winds indicate circulation in the lower atmosphere and 500 hPa zonal winds describe the dominate large-scale circulation'.

*10. Figure 5: why the calculation region for V_850hPa differently?*

**Response:**

As we explained in our response to your 'Specific Comment #2', the selection of calculating region for the index_V850 depends on the correlations shown in Fig. 5h. The strongest correlations between MDA8 $O_3$ concentrations and V850 occur in North China as well as the west region (enclosed by 35°N-45°N, 107°E - 120°E, the red rectangle in Fig. 5h).

*11. Figure 7, similar to Figure 4?*

**Response:**

Figure 4 shows the typical weather pattern for observed OPEs by composite analysis. Figure 7a-d is the same as Fig. 4d, 4e, 4b and 4a, respectively. However, Figure 7 is utilized to verify that the I_OPE, which is defined by meteorological fields only, can well represent the typical weather pattern obtained from observed OPEs. To address this concern, we have moved Figure 7 to be Figure S2 in the supplementary material.

*12. Page 9 line 21: "indicating that the GEOS-Chem model can simulate the daily variation in MDA8 O3", but the simulation not agree well to the observation in Figure 8.*

**Response:**

See our response to your 'Specific comments #3'.

*13. page 9 line 26: " The linear regression through the origin between observed and simulated MDA8 O3 has a regression coefficient of 0.96, indicating the capability of the model in simulating the MDA8 O3 concentrations." Need to be clarified.*

**Response:**

We have added Fig. S3 in the supplementary material to clarify the linear regression between observed and simulated MDA8 $O_3$ concentrations. The sentence has been revised as:

'The linear regression by the least square method through the origin between observed and simulated MDA8 $O_3$ has a regression coefficient of 0.96 (Fig. S3), indicating the capability of the model in simulating the MDA8 $O_3$ concentrations.'

[Figure]

Figure S3. The linear regression through the origin between observed and simulated MDA8 $O_3$ concentrations ($\mu g\ m^{-3}$). The black dots indicate the daily observed and simulated MDA8 $O_3$ concentrations averaged over North China from May to July in 2014-2017, and the correlation coefficient between them are given at the top-right corner. The red line indicates the regression line through the origin calculated by the least square method.

14. *Page 10, line 29: "diffusion has a small effect on the total mass of O3 in the boundary layer. However, the diffusion process is important in 30 the boundary layer, which has . . .", confused.*

**Response:**

Sorry for the confusion. We have revised the second paragraph of Sect. 5.2.1 as:

'Diffusion process in GEOS-Chem model describes the mixing in the boundary layer, which transports $O_3$ along the concentration gradient. Since $O_3$ concentrations are higher at 950 hPa to 850 hPa than at the surface (Fig. 8a), the diffusion transports $O_3$ from the upper boundary layer downwardly to the surface. As a result, the IPR analysis shows that the net mass change in $O_3$ by diffusion is negative between 950 and 850 hPa but positive at the surface (Fig. 8b). Note that the net changes in $O_3$ mass over North China by diffusion process should approximately equal to zero (Table 1) if we integrate the change in $O_3$ mass by diffusion from the surface to 850 hPa because diffusion is an internal vertical transport. The downward transport of $O_3$ by diffusion was also reported in previous IPR analyses (e.g., Khiem et al., 2010; Li et al., 2012; Tang et al., 2017).

15. *'page 10 line 31: "mass flux of diffusion for the whole boundary layer is small, it is indicated that $O_3$ aloft is transported downward to be mixed at the surface by the diffusion process", very strange explanation.*

**Response:**

See our response above (our response to #14 of your Other points).

**16.** *Page 11 line 13: "both the positive contribution at the surface and the negative contributions in the upper layers increase", ok. "indicating that more O3 is mixed from the upper levels to the surface to increase the surface O3 concentration during OPEs", why mixing/diffusion increase surface O3?*

**Response:**

As we explained in our response to your 'Other points #14', for the seasonal mean condition, $O_3$ chemical production at and above the upper boundary layer leads to higher $O_3$ concentrations there than at the surface, causing the downward transport of $O_3$ by diffusion. During OPEs, hot and sunny conditions enhance $O_3$ chemical production at and above the upper boundary layer and hence more $O_3$ is transported from the upper boundary layer to the surface.

**17.** *Page 12 line 1: "beneficial for developing the mixed layer, leading to more O3 mixed downward to the surface during OPEs", O3 produces in the boundary layer, no need to mixing down from upper layer above ABL.*

**Response:**

See our response to your 'Specific comments #5'.

**18.** *page 13 line 14: "horizontal advection, and vertical advection are the most dominant processes that lead to OPEs", but horizontal advection contributes negative mass flux? So, not lead to OPEs, but depress the development of OPEs.*

**Response:**

The 'horizontal advection' has been removed in the revised manuscript.

**References:**

Cai, W., Li, K., Liao, H., Wang, H., and Wu, L.: Weather conditions conducive to Beijing severe haze more frequent under climate change, Nature Climate Change, 7, 257-+, 10.1038/nclimate3249, 2017.

Ding, Y. H., and Chan, J. C. L.: The East Asian summer monsoon: an overview, Meteorology and Atmospheric Physics, 89, 117-142, 10.1007/s00703-005-0125-z, 2005.

Khiem, M., Ooka, R., Hayami, H., Yoshikado, H., Huang, H., and Kawamoto, Y.: Process analysis of ozone formation under different weather conditions over the Kanto region of Japan using the MM5/CMAQ modelling system, Atmospheric Environment, 44, 4463-4473, 10.1016/j.atmosenv.2010.07.038, 2010.

Li, K., Chen, L., Ying, F., White, S. J., Jang, C., Wu, X., Gao, X., Hong, S., Shen, J., Azzi, M., and Cen, K.: Meteorological and chemical impacts on ozone formation: A case study in Hangzhou, China, Atmospheric Research, 196, 40-52, 10.1016/j.atmosres.2017.06.003, 2017a.

Li, K., Jacob, D. J., Liao, H., Shen, L., Zhang, Q., and Bates, K. H.: Anthropogenic drivers of 2013-2017 trends in summer surface ozone in China, Proceedings of the National Academy of Sciences of the United States of America, 116, 422-427, 10.1073/pnas.1812168116, 2019.

Li, L., Chen, C. H., Huang, C., Huang, H. Y., Zhang, G. F., Wang, Y. J., Wang, H. L., Lou, S. R., Qiao,

L. P., Zhou, M., Chen, M. H., Chen, Y. R., Streets, D. G., Fu, J. S., and Jang, C. J.: Process analysis of regional ozone formation over the Yangtze River Delta, China using the Community Multi-scale Air Quality modeling system, Atmospheric Chemistry and Physics, 12, 10971-10987, 10.5194/acp-12-10971-2012, 2012.

Lou, S., Liao, H., and Zhu, B.: Impacts of aerosols on surface-layer ozone concentrations in China through heterogeneous reactions and changes in photolysis rates, Atmospheric Environment, 85, 123-138, 10.1016/j.atmosenv.2013.12.004, 2014.

Lou, S., Liao, H., Yang, Y., and Mu, Q.: Simulation of the interannual variations of tropospheric ozone over China: Roles of variations in meteorological parameters and anthropogenic emissions, Atmospheric Environment, 122, 839-851, 10.1016/j.atmosenv.2015.08.081, 2015.

Lu, X., Zhang, L., Chen, Y., Zhou, M., Zheng, B., Li, K., Liu, Y., Lin, J., Fu, T.-M., and Zhang, Q.: Exploring 2016-2017 surface ozone pollution over China: source contributions and meteorological influences, Atmospheric Chemistry and Physics, 19, 8339-8361, 10.5194/acp-19-8339-2019, 2019.

Ni, R., Lin, J., Yan, Y., and Lin, W.: Foreign and domestic contributions to springtime ozone over China, Atmospheric Chemistry and Physics, 18, 11447-11469, 10.5194/acp-18-11447-2018, 2018.

Pu, X., Wang, T. J., Huang, X., Melas, D., Zanis, P., Papanastasiou, D. K., and Poupkou, A.: Enhanced surface ozone during the heat wave of 2013 in Yangtze River Delta region, China, Sci Total Environ, 603-604, 807-816, 10.1016/j.scitotenv.2017.03.056, 2017.

Shu, L., Xie, M., Wang, T., Gao, D., Chen, P., Han, Y., Li, S., Zhuang, B., and Li, M.: Integrated studies of a regional ozone pollution synthetically affected by subtropical high and typhoon system in the Yangtze River Delta region, China, Atmospheric Chemistry and Physics, 16, 15801-15819, 10.5194/acp-16-15801-2016, 2016.

Su, T., Xue, F., and Zhang, H.: Simulating the intraseasonal variation of the East Asian summer monsoon by IAP AGCM4.0, Advances in Atmospheric Sciences, 31, 570-580, 10.1007/s00376-013-3029-8, 2014.

Sun, L., Xue, L., Wang, Y., Li, L., Lin, J., Ni, R., Yan, Y., Chen, L., Li, J., Zhang, Q., and Wang, W.: Impacts of meteorology and emissions on summertime surface ozone increases over central eastern China between 2003 and 2015, Atmospheric Chemistry and Physics, 19, 1455-1469, 10.5194/acp-19-1455-2019, 2019.

Tang, G., Zhu, X., Xin, J., Hu, B., Song, T., Sun, Y., Zhang, J., Wang, L., Cheng, M., Chao, N., Kong, L., Li, X., and Wang, Y.: Modelling study of boundary-layer ozone over northern China - Part I: Ozone budget in summer, Atmospheric Research, 187, 128-137, 10.1016/j.atmosres.2016.10.017, 2017.

Tie, X., Geng, F., Peng, L., Gao, W., and Zhao, C.: Measurement and modeling of O-3 variability in Shanghai, China: Application of the WRF-Chem model, Atmospheric Environment, 43, 4289-4302, 10.1016/j.atmosenv.2009.06.008, 2009.

Wang, Y., Zhang, Y., Hao, J., and Luo, M.: Seasonal and spatial variability of surface ozone over China: contributions from background and domestic pollution, Atmospheric Chemistry and Physics, 11, 351 1-3525, 10.5194/acp-11-3511-2011, 2011.

Wang, Z., Li, J., Wang, X., Pochanart, P., and Akimoto, H.: Modeling of regional high ozone episode observed at two mountain sites (Mt. Tai and Huang) in East China, Journal of Atmospheric Chemistry, 55, 253-272, 10.1007/s10874-006-9038-6, 2006.

Yang, Y., Liao, H., and Li, J.: Impacts of the East Asian summer monsoon on interannual variations of

summertime surface-layer ozone concentrations over China, Atmospheric Chemistry and Physics, 14, 6867-6879, 10.5194/acp-14-6867-2014, 2014.

Zhang, H., Wang, Y., Park, T.-W., and Deng, Y.: Quantifying the relationship between extreme air pollution events and extreme weather events, Atmospheric Research, 188, 64-79, 10.1016/j.atmosres.2016.11.010, 2017.

Zhang, Y., and Wang, Y.: Climate-driven ground-level ozone extreme in the fall over the Southeast United States, Proc Natl Acad Sci U S A, 113, 10025-10030, 10.1073/pnas.1602563113, 2016.

---

## Author Response (AR2)

We thank the reviewers for their valuable comments. We have made efforts to improve the manuscript accordingly, please find the responses to the corresponding points below.

**Two additional comments:**

1) *Page 10, line8-10:"simulated MDA8 O3 has a regression coefficient of 0.96 (Fig. S3), indicating the capability of the model in simulating the MDA8 O3 concentrations", regression coefficient close to 1, not necessarily indicating the model perform good, if the data is very scattered.*

**Response:**

The sentences have been revised as follows:

'Compared to observed MDA8 $O_3$ concentrations, the simulated concentrations have a mean bias (MB) (normalized mean bias (NMB)) of 2.4 µg m$^{-3}$ (1.7%) in 2014, 6.7 µg m$^{-3}$ (4.9%) in 2015, 1.8 µg m$^{-3}$ (1.2%) in 2016, and -12.5 µg m$^{-3}$ (-8.4%) in 2017, indicating the capability of the model in simulating the MDA8 $O_3$ concentrations. For all the data samples in May-July of 2014-2017, the observed MDA8 $O_3$ concentration averaged over North China is 146.8 µg m$^{-3}$, and the simulated mean value is also 146.8 µg m$^{-3}$. The linear regression between the observed and simulated MDA8 $O_3$ by the least-square method through the origin has a regression coefficient of 0.96 (Fig. S3).'

2) *Page 11 line 19-21:"Note that the net changes in O3 mass over North China by diffusion process should approximately equal to zero (Table 1) if we integrate the change in O3 mass by diffusion from the surface to 850 hPa because diffusion is an internal vertical transport", This is not in good expression. What is internal vertical transport? Diffusion just redistributes the O3 mass, neither creation, nor destruction.*

**Response:**

Thanks for the comment. We have revised this sentence as:

[revised manuscript text omitted]